# Improved Lower Bounds for First-order Stochastic Non-convex Optimization under Markov Sampling

**Zhenyu Sun**[1] **Ermin Wei**[1,2]

## Abstract

Unlike its vanilla counterpart with i.i.d. samples, stochastic optimization with Markovian sampling allows the sampling scheme following a Markov chain. This problem encompasses various applications that range from asynchronous distributed optimization to reinforcement learning. In this work, we lower bound the sample complexity of finding $\epsilon$-approximate critical solutions for any first-order methods when sampling is Markovian. We show that for samples drawn from stationary Markov processes with countable state space, any algorithm that accesses smooth, non-convex functions through queries to a stochastic gradient oracle, requires at least $\Omega(\epsilon^{-4})$ samples. Moreover, for finite Markov chains, we show a $\Omega(\epsilon^{-2})$ lower bound and propose a new algorithm, called MaC-SAGE, that is proved to (nearly) match our lower bound.

## 1. Introduction

Stochastic optimization methods have become integral to a wide array of machine learning and statistical modeling tasks, with applications ranging from large-scale data analysis to reinforcement learning and control. Due to the prosperity in deep learning and large models, first-order optimization methods, meaning only gradient information is leveraged for algorithm design, stand out due to their advantages of implementability and computational efficiency (Achiam et al., 2023; Zhao et al., 2023). In many real-world scenarios, however, data do not arrive independently and identically distributed (i.i.d.). Instead, they are generated by underlying dynamical processes exhibiting temporal

or sequential dependencies, naturally modeled by Markov chains. For instance, in reinforcement learning, an agent typically collects data by interacting with an environment governed by a Markov decision process (MDP) (Sutton, 2018), leading to correlated samples. In online recommendation systems, user feedback often arrives sequentially in a Markovian fashion, where each user's response depends on their previous engagements (Afsar et al., 2022). In time-series analysis, measurements such as sensor data or stock prices reflect underlying Markovian transitions (Hamilton, 2020; Esling & Agon, 2012). In the context of Bayesian inference, Markov chain Monte Carlo (MCMC) methods generate correlated samples to approximate posterior distributions (Brooks, 1998; Nemeth & Fearnhead, 2021), which in turn are used for high-dimensional parameter estimation. Moreover, in the language models, transformers generate next-token predictions via a Markov chain (Makkuva et al., 2024). The above-mentioned practical applications highlight the importance of generalizing analysis from the i.i.d. sampling to Markov sampling. Over the last few years, a growing body of literature has begun addressing questions involving Markov sampling given specific contexts, such as reinforcement learning (Bhandari et al., 2018; Zhang et al., 2021), distributed optimization (Sun et al., 2023; Even et al., 2024), and federated learning (Sun et al., 2024). This triggers the need for a unified theory advance that captures the intrinsic difficulty of dealing with Markovian dynamics.

Compared to the well-studied i.i.d. setting, analyses of first-order optimization methods under Markov sampling are faced with novel and substantial challenges posed by correlated data. In the i.i.d. scenario, each data point is drawn independently from the same distribution and hence renders unbiased estimate of stochastic gradient, which allows straightforward variance bounds and concentration inequalities to be applied (Nemirovski et al., 2009; Lan, 2012; Li & Liu, 2022). In contrast, when samples follow a Markov chain, neighboring data observations are correlated, which complicates both the bias and variance characterizations of stochastic gradient estimates. In fact, due to the Markovian property, the gradient estimate is biased, prohibit direct generalization of analyses from the i.i.d. case (Bhandari et al., 2018; Even, 2023; Roy et al., 2022; Sun et al., 2023; Kim et al., 2022).

[1]Department of Electrical and Computer Engineering, Northwestern University, Evanston, IL, USA [2]Department of Industrial Engineering and Management Science, Northwestern University, Evanston, IL, USA. Correspondence to: Zhenyu Sun <zhenyusun2026@u.northwestern.edu>.

*Proceedings of the 42$^{nd}$ International Conference on Machine Learning*, Vancouver, Canada. PMLR 267, 2025. Copyright 2025 by the author(s).

Moreover, existing literature investigating stochastic optimization under Markov sampling mainly focuses on deriving various convergence and sample complexity upper bounds for different algorithms. For example, recently (Even, 2023) provides the sample complexity upper bound for vanilla SGD under the Markov sampling scheme. (Dorfman & Levy, 2022) proposes sophisticated variant of SGD to guarantee robustness in hyperparameter selection. (Beznosikov et al., 2024) further establishes the convergence analysis for accelerating first-order methods and generalizes them to variational inequalities. However, the lower bound results are still lacking, especially when the objective function is non-convex, even they are fruitful for (strongly) convex case and for the i.i.d. case. Therefore, in this paper we aim to bridge such research gap by establishing the lower bounds on sample complexities of first-order methods that solve stochastic non-convex optimization problems when data samples are Markovian. Our main contributions are summarized as follows:

- We provide the algorithm-independent sample complexity lower bound for any first-order methods of stochastic non-convex optimization problems, given data samples are generated by a countable-state stationary Markov chain. Our lower bound shows a complexity with the order of $\epsilon^{-4}$, which matches the upper bound of algorithm MAG provided in (Dorfman & Levy, 2022; Beznosikov et al., 2024).

- We further restrict on the case of finite-state Markov chains and show an $\epsilon^{-2}$ lower bound on the sample complexity. The bound is not contradictory to the bound for the countable-state case, as we are search under different function and oracle classes.

- We then propose a new algorithm, which is called MaC-SAGE, when the Markov chain is finite-state. The convergence analysis is provided for MaC-SAGE, which indicates nearly same order of $\epsilon^{-2}$, hence demonstrating the (near) min-max optimality of the proposed algorithm.

**Sample complexity of stochastic optimization.** The sample complexity analysis of first-order stochastic optimization has thrived since last two decades. Lower bound results are undoubtedly crucial, as these bounds provide a fundamental limit on how efficiently any first-order algorithm can learn. Typically when full batch of samples are used for algorithm design, the lower bound of $\Omega(\epsilon^{-2})$ is established for smooth non-convex functions (Carmon et al., 2020) and algorithm Gradient Descent matches this lower bound (Ghadimi & Lan, 2013). When samples are assumed to be i.i.d., $\Omega(\epsilon^{-2})$ and $\Omega(\epsilon^{-4})$ (which improves to $\Omega(\epsilon^{-3})$ if the objective is assumed mean-square smooth (Arjevani et al., 2023)) lower bounds are provided for convex (Agarwal et al., 2009) and

non-convex objectives (Arjevani et al., 2023), respectively, and SGD is shown to have matched upper bounds for both cases (Foster et al., 2019; Ghadimi & Lan, 2013). For the Markovian sampling scheme, denoting $\tau$ as the mixing or hitting time of the Markov chain, (Beznosikov et al., 2024) shows a bound with $\Omega(\tau \log(\epsilon^{-1}))$ for strongly convex functions and (Duchi et al., 2012) provides a bound of $\Omega(\tau \epsilon^{-2})$ for the convex case which is proven to match the upper bound of SGD (Duchi et al., 2012). Recently, (Even, 2023) establishes a loose lower bound $\Omega(\tau \epsilon^{-1})$ for non-convex functions, while the best-known upper bound for this case is $\mathcal{O}(\tau \epsilon^{-4})$. Table 1 compares various lower bounds given different function and sampling assumptions.

## 2. Problem Formulation

Consider the general stochastic optimization problem,

$$\min_x \quad F(x) := \mathbb{E}_{s \sim \Pi}[f(x; s)] \tag{1}$$

where $s \in \mathcal{S}$ for $\mathcal{S}$ being the support, and $\Pi$ denotes some unknown underlying distribution. In this paper we focus on the Markovian case, i.e., we assume that the samples $\{s_t\}_{t=0}^{\infty}$ form a sequence generated by some underlying Markov chain with its stationary distribution being $\Pi$. Moreover we focus on countable-state Markov chains, meaning the state spaces are countable but may not be finite. Note that the Markovian setting reduces to the i.i.d. setting by decoupling the dependence across time.

Since exactly solving (1) is NP-hard (Hillar & Lim, 2013), by restricting to first-order methods, we search for an $\epsilon$-approximate critical solution, which is widely adopted by literature (Carmon et al., 2020; Arjevani et al., 2023; Beznosikov et al., 2024) of $F(x)$ defined in (1). In particular, given differentiable function $F : \mathbb{R}^d \to \mathbb{R}$, our goal is to find some $x$ such that

$$\|\nabla F(x)\| \leq \epsilon$$

for any $\epsilon > 0$.

### 2.1. An Example: Temporal-difference Learning

To illustrate the importance of our problem, we show that the temporal-difference (TD) learning algorithm can be viewed a special case that iteratively searching for an $\epsilon$-approximate critical solution of problem (1). Particularly, for the TD learning, we aim to (approximately) learn the value function defined by

$$V^*(s) := \mathbb{E}\left[\sum_{t=0}^{\infty} \gamma^t r(s_t, s_{t+1}) \mid s_0 = s\right]$$

where $r(s_t, s_{t+1})$ is the reward function and $s_{t+1} \sim P(\cdot \mid s_t)$ is drawn from some unknown stationary Markov chain $P$.

*Table 1.* Sample complexity lower bounds for stochastic optimization with smooth objectives. "full" means the objective function is directly used, corresponding to deterministic case; for convex case, the sample complexity measure metric is $F(x) - F^* \leq \epsilon$; for non-convex case, the complexity measure metric is $\|\nabla F(x)\| \leq \epsilon$; $\tau$ represents the hitting/mixing time.

| Reference | Convexity | Sampling | Lower bound |
| --- | --- | --- | --- |
| (Nesterov, 2013) | strongly convex | full[1] | $\Omega(\log(\epsilon^{-1}))$ |
| (Nesterov, 2013) | convex | full | $\Omega(\epsilon^{-0.5})$ |
| (Carmon et al., 2020) | non-convex | full | $\Omega(\epsilon^{-2})$ |
| (Agarwal et al., 2009) | convex | i.i.d. | $\Omega(\epsilon^{-2})$ |
| (Arjevani et al., 2023) | non-convex | i.i.d. | $\Omega(\epsilon^{-4})$ |
| (Beznosikov et al., 2024) | strongly convex | Markovian | $\Omega(\tau \log \epsilon^{-1})$ |
| (Duchi et al., 2012) | convex | Markovian | $\Omega(\tau \epsilon^{-2})$ |
| (Even, 2023) | non-convex | Markovian | $\Omega(\tau \epsilon^{-1})$ |
| This work | non-convex | Markovian | $\Omega(\tau \epsilon^{-4})$ |

Assuming that the value function can be parameterized by a linear function, i.e., there exists some $\theta^*$ such that $V^*(s) = \phi(s)^T \theta^*, \forall s \in \mathcal{S}$ for the known feature mapping $\phi(\cdot)$, then the TD-learning algorithm maintains an estimate $\theta$ of $\theta^*$ by the following update:

$$\theta_{t+1} = \theta_t - \eta_t(\phi(s_t)^T \theta_t - r(s_t, s_t') - \gamma\phi(s_t')^T \theta_t)\phi(s_t)$$
$$:= \theta_t - \eta_t g(\theta_t; s_t, s_t') \qquad (2)$$

where $s_t'$ is the sample drawn from $s_t' \sim P(\cdot \mid s_t)$, $g(\theta_t; s_t, s_t') = (\phi(s_t)^T \theta_t - r(s_t, s_t') - \gamma\phi(s_t')^T \theta_t)\phi(s_t)$ and $\eta_t$ is the stepsize. Defining the augmented state $\bar{s} = (s, s') \in \mathcal{S} \times \mathcal{S}$ for which $s' \sim P(\cdot \mid s)$ and letting

$$f(\theta; \bar{s}) = \int_0^1 g(\theta_0 + u(\theta - \theta_0); s, s')^T(\theta - \theta_0)du$$

with $g(\theta; s, s')$ defined in (2), we have $\nabla f(\theta; \bar{s}) = g(\theta; s, s')$, implying that (2) is equivalent to

$$\theta_{t+1} = \theta_t - \eta_t \nabla f(\theta_t; \bar{s}_t) \qquad (3)$$

where $\{\bar{s}\}_{t=0}^{\infty}$ forms another Markov chain with augmented state space $\tilde{\mathcal{S}} = \mathcal{S} \times \mathcal{S}$. Shown by (Tsitsiklis & Van Roy, 1996; Bhandari et al., 2018) (2) asymptotically converges to the solution to the following equation in expectation:

$$\|\mathbb{E}_{s\sim\Pi, s'\sim P(\cdot|s)}[g(\theta; s, s')]\| = 0$$

where $\Pi$ is the stationary distribution corresponding to the Markov chain $P$. It then equivalently yields by (3) that the TD-learning algorithm outputs an $\epsilon$-approximation critical solution of some $F$ for which

$$\mathbb{E}_{\bar{s}=(s,s')\sim\Pi\times P(\cdot|s)}[\nabla f(\theta; \bar{s})] =: \nabla F(\theta).$$

## 2.2. Function class

Particularly, we consider all smooth functions in the following set:

$$\mathcal{F}(\Delta, L) := \big\{F : \mathbb{R}^d \to \mathbb{R} \mid F(0) - \inf_x F(x) \leq \Delta,$$
$$\|\nabla F(x) - \nabla F(y)\| \leq L\|x - y\|, \forall x, y \in \mathbb{R}^d\big\} \qquad (4)$$

where $\Delta \geq 0$ and $L > 0$ are fixed parameters. The condition $F(0) - \inf_x F(x) \leq \Delta$ on $F(0)$ can be generalized to any initial value $F(x_0)$. However, for *zero-respecting* algorithms (to be defined in Section 2.3), we have $x_0 = 0$. In particular, we consider the case where the objective $F$ is smooth and has bounded initial gap to the optimum.

## 2.3. Algorithm class

Our algorithm class is based on the flow of (Arjevani et al., 2023). We consider the following first-order algorithms such that:

- the algorithm access an unknown $F \in \mathcal{F}(\Delta, L)$ by a stochastic first-order oracle $O$;

- the oracle $O$ returns a sequence of samples $z := \{s_i\}_{i=1}^B$ ($B$ can be time-dependent) generated by a Markov chain and a mapping

$$O_F(x, \{s_i\}_{i=1}^B) := \{g(x; s_i))\}_{i=1}^B$$

where $g(x; s) := \nabla f(x; s)$ is the stochastic gradient.

- at iteration $t$, the algorithm queries a batch of $M$ points

$$x_t := (x_{t,1}, x_{t,2}, \ldots, x_{t,M});$$

- for each batch query $x_t$, $O$ responses with

$$O_F(x_t, z_t) := (O_F(x_{t,1}, z_{t,1}), \ldots, O_F(x_{t,M}, z_{t,M})),$$

  where $z_{t,i}$ is the sequence of sample drawn for $x_{t,i}$ and $z_t := \bigcup_{i=1}^M z_{t,i}$.

Then algorithm $\mathcal{A}$ consists of a sequence of measurable mappings $\{\mathcal{A}_t\}_{t=0}^\infty$ to generate a sequence of iterates $\{x_t\}_{t=0}^\infty$ satisfying the following conditions:

- the $t + 1$-th iterate is the output of $\mathcal{A}_t$ when taking all previous oracle responses as input, i.e.,

$$x_{t+1}^{\mathcal{A}[O_F]} = \mathcal{A}_t \left( O_F(x_0^{\mathcal{A}[O_F]}, z_0), \ldots, O_F(x_t^{\mathcal{A}[O_F]}, z_t) \right);$$

- Algorithm $\mathcal{A}$ is *zero-respecting*, i.e., for any $O$ and samples $z_0, z_1, \ldots$ with any $M$, it satisfies for any $t \geq -1$ and any $m \in [M]$

$$\text{support}(x_{t+1,m}^{\mathcal{A}[O_F]}) \subseteq \bigcup_{k \leq t, m' \in [M]} \text{support}(g_{k,m'}), \quad (5)$$

  where $g_{k,m'}$ is the stochastic gradient for $x_{k,m'}^{\mathcal{A}[O_F]}$ and $\text{support}(x) := \{i \in [d] : [x]_i \neq 0\}$ for $[x]_i$ being the $i$-th coordinate of $x$.

We denote $\mathbf{A}_{zr}(M)$ the class of all zero-respecting algorithms. It is worth noting that for any $\mathcal{A} \in \mathbf{A}_{zr}(M)$, $x_{0,1}^{\mathcal{A}[O_F]} = 0$ by definition.

We note that the above-mentioned algorithm class is general, which captures many existing first-order algorithms. For example, the vanilla MC-SGD (Even, 2023)

$$x_{t+1} = x_t - \eta_t g(x_t; s_t)$$

corresponds to $M = 1, B = 1, \forall t \geq 0$. For Randomized ExtraGradient (Beznosikov et al., 2024), which maintains the update by

$$x_{t+1/2} = \text{prox}_{\eta_t}(x_t - \eta_t g(x_t; s_{T_t+1}))$$
$$x_{t+1} = \text{prox}_{\eta_t}(x_{t+1/2} - \eta_t u_t)$$

where by generating $J_t \sim \text{Geom}(1/2)$

$$u_t = u_t^0 + \begin{cases} 2^{J_t}(u_t^{J_t} - u_t^{J_t-1}) & \text{if } 2^{J_t} \leq K \\ 0 & \text{otherwise} \end{cases}$$

with

$$u_t^j := 2^{-j} \sum_{i=1}^{2^j} g(x_t; s_{T_t+i+1}), \quad T_{t+1} = T_t + 1 + 2^{J_t}.$$

one can clearly see that it fits in the case of $M = 2$ and $B = 2^{J_t}$.

## 2.4. Countable-state Markov Chain

In this section, we are interested in any sampling schemes characterized by countable-state Markov chains, meaning that the state space $\mathcal{S}$ is discrete while $|\mathcal{S}| = \infty$ is allowed. Particularly, when the state space is finite, i.e., $|\mathcal{S}| < \infty$, the case reduces to that of finite-state Markov chains, which is separately analyzed in Section 4.

The class of countable-state Markov chains is parameterized by the hitting time defined as follows.

**Definition 2.1** (Hitting time). For any state $w \in \mathcal{S}$, define

$$\tau_w := \inf\{t \geq 1 \mid s_t = w\}$$

as the Markov chain firstly reaches state $w$. The hitting time $\tau_{hit}$ is defined by

$$\tau_{hit} := \max_{v,w \in \mathcal{S} \times \mathcal{S}} \mathbb{E}[\tau_w \mid s_0 = v].$$

Intuitively, the hitting time measures the maximal number of steps for which any pair of states take to transit between each other.

Then we consider the class of countable-state Markov chains for which the stationary distribution $\Pi$ exists and the hitting time $\tau_{hit}$ is upper bounded by parameter $\tau \geq 1$. We denote the chain by $P$. Specifically,

$$\mathcal{M}_s(\tau) := \left\{ P \mid \tau_{hit} \leq \tau, \lim_{t \to \infty} \mu P^t = \Pi, \forall \mu \right\} \quad (6)$$

where $\tau_{hit}$ is defined in Definition 2.1 and $\mu P^t$ represents the distribution of the chain after $t$-step transitions starting from the initial distribution $\mu$.

## 2.5. Oracle Class

Recalling that the oracle $O$ returns a sequence of stochastic gradient evaluated at each query, we place the following assumption.

**Assumption 2.2.** For any $x \in \mathbb{R}^d$, denoting $\Pi$ as the stationary distribution of the Markov chain, $\mathbb{E}_{s \sim \Pi} \|g(x; s) - \nabla F(x)\|^2 \leq \sigma^2$ for some $0 < \sigma < \infty$, and $\mathbb{E}_{s \sim \Pi}[g(x; s)] = \nabla F(x)$.

Basically, Assumption 2.2 requires 1) asymptotically unbiased gradient estimate when the Markov chain reaches its stationary distribution $\Pi$, i.e., $\mathbb{E}_{s_t \sim \Pi}[g(x; s_t)] = \nabla F(x)$; and 2) bounded variance after convergence of the chain to its stationary distribution, i.e., $\mathbb{E}_{s \sim \Pi} \|g(x; s) - \nabla F(x)\|^2 \leq \sigma^2$. This assumption becomes aligned with the bounded variance assumption of stochastic first-order methods under i.i.d. sampling by further forcing independence across samples (Ghadimi & Lan, 2013; Allen-Zhu & Hazan, 2016).

Then, the oracle class, denoted by $\mathbb{O}_s(\sigma^2, \tau)$, is that the stochastic gradient $g$ is sampled from a chain contained in $\mathcal{M}_s(\tau)$ by (6) and such that Assumption 2.2 is satisfied.

## 2.6. Sample Complexity Measure

Our result of lower bound is established in terms of the sample complexity for finding an $\epsilon$-approximate critical solution of $F$. Let $S_t(\mathcal{A}) = \bigcup_{s \le t} z_t$ be the collection of all samples utilized til time $t$ by algorithm $\mathcal{A}$.

Concretely, the sample complexity measure is defined by

$$
N_s^\epsilon(M, \Delta, L, \sigma^2, \tau)
$$
$$
:= \sup_{O \in \mathbb{O}_s(\sigma^2, \tau)} \sup_{F \in \mathcal{F}(\Delta, L)} \inf_{\mathcal{A} \in \mathbf{A}_{zr}(M)}
$$
$$
\inf \left\{ |S_T(\mathcal{A})| \ge 1 \mid \mathbb{E}\|\nabla F(x_{T,1}^{\mathcal{A}[O_F]})\| \le \epsilon \right\}. \quad (7)
$$

When $N_s^\epsilon(M, \Delta, L, \sigma^2, \tau)$ is lower bounded by $N_T$, i.e., $N_s^\epsilon(M, \Delta, L, \sigma^2, \tau) \ge N_T$ with $N_T$ denoting all collected samples up to time $T$, it indicates that there exists some stationary Markov sampling process $P$ with bounded hitting time and an oracle $O \in \mathbb{O}_s(\sigma^2, \tau)$ such that for any $\mathcal{A} \in \mathbf{A}_{zr}(M)$ there exists $F \in \mathcal{F}(\Delta, L)$ for which $\mathbb{E}\|\nabla F(x_{T,1}^{\mathcal{A}[O_F]})\| > \epsilon$, where the expectation is taken over randomness in $\mathcal{A}$ and $O$. In other words, at least $N_T$ number of samples must be required to (possibly) achieve an $\epsilon$-approximate critical solution for any first-order algorithm.

## 3. Improved Lower Bound

In this section, we show our main result on the lower bound of sample complexity for stochastic non-convex optimization under Markov sampling. The result is algorithm-independent, implying that all first-order methods that are zero-respecting take at least such samples to reach an $\epsilon$-approximate critical point of the non-convex objective function.

When the sampling process is characterized by a countable-state stationary Markov chains, we show the following sample complexity lower bound.

**Theorem 3.1.** *Considering the samples are generated by stationary Markov chains in $\mathcal{M}_s(\tau)$, there exist numerical constants $c_1, c_2 > 0$ such that for any $M, L, \Delta, \sigma, \tau > 0$,*

$$
N_s^\epsilon(M, \Delta, L, \sigma^2, \tau) = \Omega\left( \frac{\tau \sigma^2}{\epsilon^2} + \frac{\tau \sigma^2}{\epsilon^4} \min\left\{ c_1 \sigma^2, c_2 L \Delta \right\} \right).
$$

*Remark* 3.2. Note that the extreme case $\tau = 1$ corresponds to the i.i.d. sampling case. To see this, recalling the definition of hitting time $\tau = 1$ indicates exactly one step is taken transiting from one state to any other, which then implies the samples are drawn exactly from the stationary distribution $\Pi$ and there is no time dependence across samples drawn at different time steps, hence reducing to i.i.d. case. Thus, when $\sigma^2 \succeq L\Delta$ our lower bound result is aligned with the bound $\Omega\left( \frac{L\Delta}{\epsilon^2} + \frac{\sigma^2 L \Delta}{\epsilon^4} \right)$ provided in (Arjevani et al., 2023). Moreover, noting that both lower bounds reduce

to $\Omega\left( \frac{\tau L \Delta}{\epsilon^2} + \frac{\tau \sigma^2 L \Delta}{\epsilon^4} \right)$ if $\sigma^2 \succeq L\Delta$, it matches the best-known upper bound for stationary Markov chains, which is $\mathcal{O}\left( \frac{\tau L \Delta}{\epsilon^2} + \frac{\tau \sigma^2 L \Delta}{\epsilon^4} \right)$ (Beznosikov et al., 2024).

## 4. Min-max Optimality for Finite Stationary Markov Chains

In this section, we restrict on the case of finite-state stationary Markov chains, where we further assume that the state space of stationary Markov chains is finite, i.e., $|\mathcal{S}| < \infty$. Then, we can further show a $\Omega(\tau \epsilon^{-2})$ sample complexity lower bound, which is followed by a new proposed algorithm (MaC-SAGE) with $\tilde{\mathcal{O}}(\max\{\tau, \tau_{mix}\}\epsilon^{-2})$ sample complexity ($\tau_{mix}$ is the mixing time defined in Defintion B.1), indicating the nearly min-max optimality of our proposed algorithm.

We firstly define the class of finite-state Markov chains:

$$
\mathcal{M}_{s,fi}(\tau) := \{P \mid P \in \mathcal{M}_s(\tau), |\mathcal{S}| < \infty\}. \quad (8)
$$

Our subsequent analysis is established on the oracle class denoted by $\mathbb{O}_{s,fi}(\sigma^2, \tau)$, which requires the sampled stochastic gradients are drawn from a stationary Markov chain $P \in \mathcal{M}_{s,fi}(\tau)$ and also satisfy bounded noise assumption given as follows:

**Assumption 4.1.** For any $x \in \mathbb{R}^d$ and any $t \ge 0$, $\|g(x; s_t) - \nabla F(x)\|^2 \le \sigma^2$ for $\sigma > 0$ and if $s_t \sim \Pi, \forall t$, $\mathbb{E}_{s_t \sim \Pi}[g(x; s_t)] = \nabla F(x)$.

Moreover, given the finiteness of the state space, we consider objective functions are in the following class

$$
\mathcal{F}'(\Delta, L) := \left\{ F : \mathbb{R}^d \to \mathbb{R} \mid F(0) - \inf_x F(x) \le \Delta, \right.
$$
$$
\left. \|\nabla f(x; s) - \nabla f(y; s)\| \le L\|x - y\|, \forall x, y \in \mathbb{R}^d, \forall s \in \mathcal{S} \right\}
$$
$$
\quad (9)
$$

and note that $\mathcal{F}' \subset \mathcal{F}$ by further place smoothness pointwisely on every $f(\cdot; s)$, which is commonly used in literature (Even, 2023). Accordingly, the sample complexity measure is given by

$$
N_{s,fi}^\epsilon(M, \Delta, L, \sigma^2, \tau)
$$
$$
:= \sup_{O \in \mathbb{O}_{s,fi}(\sigma^2, \tau)} \sup_{F \in \mathcal{F}'(\Delta, L)} \inf_{\mathcal{A} \in \mathbf{A}_{zr}(M)}
$$
$$
\inf \left\{ |S_T(\mathcal{A})| \mid \mathbb{E}\|\nabla F(x_{T,1}^{\mathcal{A}[O_F]})\| \le \epsilon \right\}, \quad (10)
$$

based on which we provide the following lower bound of sample complexity for finite-state stationary Markov chains:

**Theorem 4.2.** *Considering the samples are generated by finite-state Markov chains in $\mathcal{M}_{s,fi}(\tau)$, there exist numerical constants $c_3, c_4 > 0$ such that for any $M, L, \Delta, \sigma, \tau > 0$,*

$$
N_{s,fi}^\epsilon(M, \Delta, L, \sigma^2, \tau) = \Omega\left( \frac{\tau}{\epsilon^2} \min\left\{ c_3 \sigma^2, c_4 L \Delta \right\} \right).
$$

Note that the results shown in Theorems 3.1 and 4.2 are not contradictory, in the sense that $\mathbb{O}_{s,f} \subset \mathbb{O}_s$ by $\mathcal{M}_{s,fi}(\tau) \subset \mathcal{M}_s(\tau), \mathcal{F}' \subset \mathcal{F}$ and Assumption 4.1 implying Assumption 2.2. Therefore, lower bounding $N_s^\epsilon$ allows us to search for a "hard" example in a broader space, compared to doing so for $N_{s,f}^\epsilon$. In other words, the "hard" example in $\mathbb{O}_s$ that realizes $\Omega(\tau\epsilon^{-4})$ of Theorem 3.1 is more special than the one to realize $\Omega(\tau\epsilon^{-2})$ of Theorem 4.2 and is probably unrealizable when restricting on the class $\mathbb{O}_{s,fi}$. Technically speaking, finding such an example for $\mathbb{O}_s$ needs more sophisticated construction than for $\mathbb{O}_{s,fi}$.

A natural question following is then to ask that

*Is the lower bound in Theorem 4.2 tight enough?*

If we expect a positive answer to this question, an algorithm should be designed such that it has an orderly-same sample complexity upper bound, i.e., we need to show that $N_{s,fi}^\epsilon(M, \Delta, L, \sigma^2, \tau) = \mathcal{O}(\tau\epsilon^{-2})$. In fact, we propose a new algorithm, called MaC-SAGE (summarized in Algorithm 1), which achieves a nearly-same order of $\tau\epsilon^{-2}$.

As now the state space $\mathcal{S}$ of the Markov chain is finite, to state our algorithm we simply set $|\mathcal{S}| = n$ and denote $s_t \in \mathcal{S}$ as the state visited at time $t$. Further, we use $s_t = i$ to indicate the $i$-th state is visited at time $t$, hence correspondingly $\nabla f(x_t, s_t) := \nabla f_i(x_t)$. In Algorithm 1 only one sample is drawn from the underlying Markov chain at each time iteration, which returns the corresponding stochastic gradient evaluated at the current query $x_t$. Simultaneously a vector $y_t$ is maintained to track the number of occurrences of each state, serving as a role to reweigh the contribution of the gradient given by each state. Then $h_t$ is designed to dynamically track the latest gradient information provided by every state, which is then combined together with $y_t$ to incorporate corrected gradient into $G_t$. Defining $a_i(t) := \sup\{l \mid l \le t, s_l = i\}$, it is straightforward to observe that Algorithm 1 can be rewritten as follows:

$$x_{t+1} = x_t - \gamma_t G_t$$

where

$$y_t^i = \frac{1}{t+1}\sum_{l=0}^{t} \mathbb{1}_{s_l=i}, \ \forall i \in [n],$$

$$G_t = \sum_{i=1}^{n} y_t^i \nabla f_i(x_{a_i(t)}).$$

which is similar to the classical variance-reduced algorithm SAG except for the weight $y_t^i$. As a matter of fact, Algorithm 1 is inspired by SAG (Schmidt et al., 2017) by further introducing $y_t$ as an estimator of the stationary distribution $\Pi$. Intuitively, when $\Pi$ is known, SAG would effectively reduce the variance and thus speedup the convergence rate as now

---

**Algorithm 1** **Ma**rkov-Chain **S**tochastic **A**verage **G**radient with **E**stimation (MaC-SAGE)

1: **Input:** Initialize $x_0$, $y_{-1} = 0_n$, $h_{-1}^i = 0_d, \forall i \in [n]$, $G_{-1} = 0_d$, stepsizes $\{\gamma_t\}$.
2: **for** $t = 0, 1, \ldots, T-1$ **do**
3:     Sample state $s_t = i$ from the underlying Markov chain, i.e., $f(\cdot; s_t) = f_i(\cdot)$.
4:     Update $y_t = (y_t^1, \ldots, y_t^n)$ as

$$y_t^j = \frac{t}{t+1}y_{t-1}^j + \frac{1}{t+1}\mathbb{1}_{j=i}, \ \forall j \in [n]. \quad (11)$$

5:     Calculate

$$G_t = G_{t-1} - y_{t-1}^i h_{t-1}^i + y_t^i \nabla f_i(x_t).$$

6:     Update $h_t = (h_t^1, \ldots, h_t^n)$ as

$$h_t^j = h_{t-1}^j + \mathbb{1}_{j=i}(\nabla f_i(x_t) - h_{t-1}^i), \ \forall j \in [n].$$

7:     Update $x_{t+1} = x_t - \gamma_t G_t$.
8: **end for**

---

$F$ is a (weighted) finite sum. Therefore, one may expect that if the estimator $y_t$ of $\Pi$ is asymptotically unbiased, meaning $\lim_{t\to\infty} y_t = \Pi$ and if the rate of $y_t$ converging to $\Pi$ is no slower than the rate of SAG (which is $\mathcal{O}(T^{-0.5})$), the sample complexity would be $\mathcal{O}(\epsilon^{-2})$ (up to some logrithmic factors). This intuitive result is summarized in the following lemma (which is restated by Corollary B.4 in Appendix B).

**Lemma 4.3.** *Suppose $y_t = (y_t^1, \ldots, y_t^n)$ with $y_{-1} = 0_n$ is updated as (11) in Algorithm 1. Then, we have for any $t \ge 1$*

$$\mathbb{E}\|y_t - \Pi\|^2 = \mathcal{O}\left(\frac{\tau_{mix}}{t}\right).$$

*where $\tau_{mix}$ is the mixing time of the chain defined in Definition B.1.*

Formally, the convergence result of MaC-SAGE is presented in the following. The proof is shown in Appendix B.

**Theorem 4.4.** *Let $F \in \mathcal{F}'(\Delta, L)$ defined in (9) with $\Delta, L > 0$. Suppose Assumption 4.1 is satisfied. Then, for any finite-state stationary Markov chain contained in $\mathcal{M}_{s,fi}(\tau)$ defined in (8) with $\tau > 0$, the trajectory $\{x_t\}_{t=0}^{T}$ generated by MaC-SAGE (Algorithm 1) satisfies*

$$\mathbb{E}\left[\min_{t<T}\|\nabla F(x_t)\|^2\right] = \mathcal{O}\left(\frac{\tau L \Delta}{T}\right) + \tilde{\mathcal{O}}\left(\frac{\tilde{\tau}\sigma^2}{T}\right),$$

*where $\tilde{\tau} = \max\{\tau, \tau_{mix}\}$ where $\tau_{mix}$ denotes the mixing time of the chain defined by Definition B.1.*

It is straightforward to observe from Theorem 4.4 that the sample complexity of MaC-SAGE to achieve an $\epsilon$-approximate critical solution is $\tilde{\mathcal{O}}(\tilde{\tau}\epsilon^{-2})$, implying its (near) optimality as it (nearly) matches the lower bound provided by Theorem 4.2.

# 5. Proof Idea of the Lower Bounds

In this section we present the proof idea of how we obtain the sample complexity lower bounds for both finite-state and countable-state stationary Markov chains. We first clarify the proof sketch by focusing on the case where $B = 1$, i.e., only one sample is drawn from the underlying Markov chain by the algorithm. Then, we generalize it to the case of $B \geq 1$ which can also be time-dependent. Full proofs are presented in Appendix A.

The core technique inspired by (Arjevani et al., 2023) is to construct a "hard" function $F$ with $f(\cdot; s)$ supported on each state of a Markov chain lying in the required class such that the gradient norm, $\|\nabla F(x)\|$, is small only if each coordinate of $x$ has a large enough absolute value. We use the progress function to mathematically evaluate the largest coordinate whose absolute value is larger than some nonnegative scalar $\alpha$, i.e.,

$$\text{prog}_\alpha(x) := \max\{k \geq 1 \mid |[x]_k| > \alpha\}$$

where $[x]_k$ represents the $k$-th coordinate of $x$. We set $\text{prog}_\alpha(x) = 0$ if $|[x]_k| \leq \alpha, \forall k \in [d]$. Then the task of finding an $\epsilon$-approximate critical solution is equivalently transformed to finding a solution $x$ whose coordinate progress is high. Formally it is stated by the following lemma.

**Lemma 5.1.** *There exists some* $F^* \in \mathcal{F}'(\mathcal{O}(\Delta\epsilon^2 d), L) \subset \mathcal{F}(\mathcal{O}(\Delta\epsilon^2 d), L)$ *such that* $\|\nabla F^*(x)\| > \epsilon, \forall \epsilon > 0$ *if* $\text{prog}_0(x) < d$.

Indicated by Lemma 5.1 ensuring $\|\nabla F^*(x)\| \leq \epsilon$ requires all coordinates of $x$ to be nonzero. Then for the case of finite-state Markov chains, we construct a chain with its hitting time upper bounded by $\tau$ and lower bounded by $\Omega(\tau)$ and design $g^*(x; s)$ such that Assumption 4.1 is satisfied and

$$
\begin{aligned}
i. \quad &\text{prog}_0(g^*(x; s)) \leq \text{prog}_0(x) + 1 \\
&\text{only if } \text{prog}_0(x) \text{ is even and } s = v^* \quad (12) \\
ii. \quad &\text{prog}_0(g^*(x; s)) \leq \text{prog}_0(x) + 1 \\
&\text{only if } \text{prog}_0(x) \text{ is odd and } s = w^* \quad (13)
\end{aligned}
$$

and otherwise $\text{prog}_0(g^*(x; s)) \leq \text{prog}_0(x), \forall s \notin \{v^*, w^*\}$. where at least $\Omega(\tau)$ number of state transitions are taken for transiting from state $v^*$ to state $w^*$ and vice versa. See Figure 1 for a visualization. With conditions (12),(13) holding we obtain that at least $\Omega(\tau d)$ iterations (if only one sample is used every iteration, i.e., $B = 1$) are required for any algorithm (which is zero-respecting) to output a solution such

that all its coordinates are nonzero, which then combines with Lemma 5.1 to guarantee the sample complexity lower bound $\Omega(\tau\epsilon^{-2})$ as shown by Theorem 4.2 by further setting $d = \Omega(\epsilon^{-2})$ (since we have to guarantee $\mathcal{O}(\Delta\epsilon^2 d) = \Delta$).

To get the lower bound for countable-state Markov chains, we modify conditions (12) and (13) such that they hold probabilistically. This can be done by splitting $v^*$ and $w^*$ into two substates, respectively, where each substate is sampled with some probability $q > 0$. Specifically, for instance if state $v^*$ is sampled there is with probability $q$ that condition (12) becomes true, similarly for the case $w^*$ is sampled. Figure 2 depicts a concrete construction of the chain. Thus, similar to the construction for the finite-state Markov chains we show the following result.

**Lemma 5.2.** *For any* $q \in (0, 1)$ *and any zero-respecting algorithm* $\mathcal{A} \in \mathbf{A}_{zr}$, *there exist a countable-state stationary Markov chain contained in* $\mathcal{M}_s(\tau)$ *and some* $F^* \in \mathcal{F}'(\mathcal{O}(\Delta d\epsilon^2), L)$ *with* $g^*(x; s)$ *satisfying* $\nabla F^*(x) = \mathbb{E}_{s \sim \Pi}[g^*(x; s)]$ *and* $\mathbb{E}\|g^*(x; s_t) - \nabla F^*(x)\|^2 \leq \mathcal{O}(\sigma^2\epsilon^2/q), \forall t \geq 0$ *such that for any* $0 < \delta < 1$, *with probability at least* $1 - \delta$

$$\max_{m \in [M]} \max_{s \leq t} \text{prog}_0(x_{s,m}^{\mathcal{A}[O_F]}) < d, \quad \forall t \leq \frac{\tau(d - \log\delta^{-1})}{4q}.$$

In fact the constructive function $F^*$ in Lemma 5.2 coincides with the one in Lemma 5.1, which then implies that at least $\Omega(\tau d/q)$ iterations are needed to guarantee an $\epsilon$-approximate critical solution output by any algorithm and hence $\Omega(\tau d/q)$ samples (due to $B = 1$). Finally setting $d = \Omega(\epsilon^{-2})$ and $q = \mathcal{O}(\epsilon^2)$ concludes the lower bound $\Omega(\tau\epsilon^{-4})$ shown by Theorem 3.1.

Note that the above proof derivations are established on the precondition when $B = 1$ by which we are able to directly obtain the sample complexity bounds through the iteration complexity analysis, since the iteration complexity is the same as the sample complexity. To generalize our results to $B \geq 1$, we present the following result.

**Lemma 5.3.** *There exist a Markov chain in* (6) *(or* (8)*) and some functions* $F^*, g^*$ *satisfying corresponding conditions in Lemma 5.2 (or Lemma 5.1) such that for any zero-respecting algorithm* $\mathcal{A}_{zr}$ *with* $B \geq 1$, *there is a zero-respecting algorithm* $\mathcal{A}_{zr}^*$ *with* $B = 1$ *for which the following holds: for any* $t \geq 0$ *if* $\max_{m \in [M]} \max_{s \leq t} \text{prog}_0(x_{s,m}^{\mathcal{A}^*[O_F]}) \leq k$, *then* $\max_{m \in [M]} \max_{s \leq t} \text{prog}_0(x_{s,m}^{\mathcal{A}[O_F]}) \leq k, \forall 0 \leq k \leq d$.

The above lemma indicates that we can always find an algorithm that only draws one sample per iteration to achieve no worse progress in its update than other algorithms that access multiple samples per iteration. In other words, combining with Lemmas 5.1 and 5.2 yields that accessing multiple samples every iteration has no benefit on improving the

sample complexity for the algorithm, which hence implies the lower bounds that holds for $B = 1$ also holds for $B \geq 1$.

## 6. Conclusion

In this paper, we study the sample complexity of general first-order stochastic non-convex optimization problems. Unlike the conventional i.i.d. sampling, we focus on the case where data samples and stochastic gradient estimates are generated by an unknown Markov chain, which introduces additional data correlation and hence non-trivial analysis difficulties. Due to the lack of sample complexity lower bound results and the gap to the best-known upper bound, we provide an improved complexity lower bound with the order of $\epsilon^{-4}$ for Markov chains with countable states, which then matches the best-known upper bound. Moreover, we establish an $\epsilon^{-2}$ lower bound for finite-state Markov chains. Finally, we propose algorithm MaC-SAGE such that its sample complexity upper bound nearly matches our lower bound, implying its near min-max optimality and the tightness of the lower bound.

## Acknowledgements

This work is supported by NSF awards CNS-2030251, ECCS-2216970, and CMMI-2024774.

## Impact Statement

This paper presents work whose goal is to advance theoretically understanding stochastic optimization. There are many potential societal consequences of our work, none which we feel must be specifically highlighted here.

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

# A. Proofs of Lower bounds

## A.1. Construction of Markov chains

We construct Markov chains which are finite-state contained in $\mathcal{M}_{s,fi}(\tau)$ and countable-state contained in $\mathcal{M}_s(\tau)$. We take the countable-state Markov chain case as an example. The construction for finite-state Markov chains are similar. Without loss of generality we assume that $\tau$ is even.

First, we pick a countable-state Markov chain $P'$ with state space $\mathcal{S}'$ (different from $\mathcal{S}$) such that its hitting time is $\lambda\tau$ for some $\lambda \in (0, 1/2)$, which can be always done since $\mathcal{M}_s(\frac{1}{2}\tau) \neq \emptyset$. Then we choose two states $s^*, w^*$ that realizes the hitting time, i.e.,

$$(s^*, w^*) \in \arg \max_{(v,w) \in \mathcal{S} \times \mathcal{S}} \mathbb{E}[\tau_w \mid s_0 = v], \ \ s.t. \ \max_{(v,w) \in \mathcal{S}' \times \mathcal{S}'} \mathbb{E}[\tau_w \mid s_0 = v] = \frac{1}{2}\tau.$$

Next we append a number of additional states $\mathcal{S} \setminus \mathcal{S}'$ into the chain $P'$ to form a new chain $P^*$ such that the state space of $P^*$ is $\mathcal{S}$. Moreover the way we append additional states satisfies that 1) there is a state $v^* \in \mathcal{S} \setminus \mathcal{S}'$ for which at least $\frac{1}{4}\tau$ steps are needed to transit between $v^*$ and $s^*$; 2) any state in $\mathcal{S} \setminus \mathcal{S}'$ must transit to state $s^*$ before transiting to other states in $\mathcal{S}'$. One specific construction of the appended chain is directed cyclic chain with self-loops, where a straightforward calculation gives its hitting time is $\tau/2$. Thus we guarantee that the new constructed chain $P^*$ has hitting time to be $\tau$. The following figure (Figure 1) illustrates a concrete example of the construction satisfying the above-mentioned requirements.

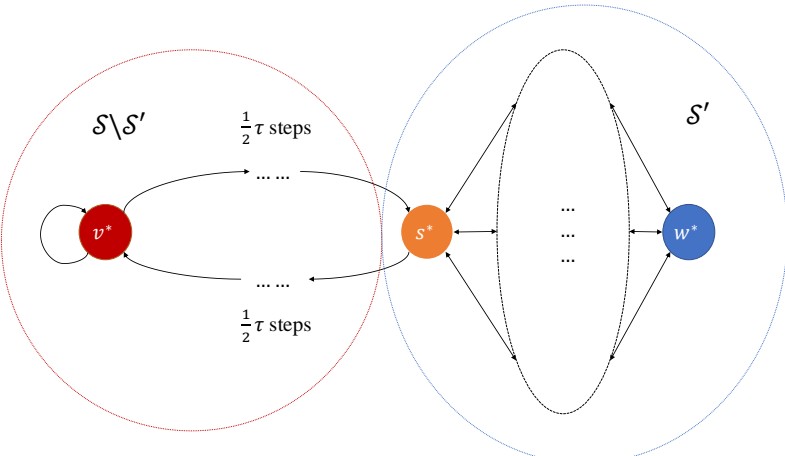

*Figure 1.* Construction of Markov chain $P^*$ with bounded hitting time

## A.2. Construction of the function $F$

Now we construct a "hard" function that is difficult for any first-order algorithm to search for the critical point. Specifically we consider the following two functions

$$h_1(x) = -\psi(1)\phi([x]_1) + \sum_{i=1}^{\lfloor d/2 \rfloor - 1} (\psi(-[x]_{2i})\phi(-[x]_{2i+1}) - \psi([x]_{2i})\phi([x]_{2i+1})) \tag{14}$$

$$h_2(x) = \sum_{i=1}^{\lfloor d/2 \rfloor} (\psi(-[x]_{2i-1})\phi(-[x]_{2i}) - \psi([x]_{2i-1})\phi([x]_{2i})) \tag{15}$$

where

$$\psi(u) = \begin{cases} 0 & , \ u \leq \frac{1}{2} \\ \exp\left(1 - \frac{1}{(2u-1)^2}\right) & , \ u > \frac{1}{2} \end{cases}$$

and

$$\phi(u) = \sqrt{e} \int_{-\infty}^{u} e^{-\frac{t^2}{2}} dt$$

with $u \in \mathbb{R}$.

We denote $\pi_s$ as the corresponding probability of state $s$ of the stationary distribution $\Pi$. Then, given the Markov chain constructed above, we know that at least $\frac{1}{2}\tau$ steps are required to take transiting from $v^*$ to $w^*$ and vice versa. Then, we construct function $F$ and $f(\cdot; s)$ such that

$$f(x; s) = \begin{cases} h_1(x), & \text{if } s = v^* \\ h_2(x), & \text{if } s = w^* \\ 0, & \text{otherwise} \end{cases} \tag{16}$$

and $F(x) = \pi_{v^*} h_1(x) + \pi_{w^*} h_2(x)$. For any $x$ and $i \geq 0$ define $x_{\leq i} := ([x]_1, \ldots, [x]_i, 0, \ldots, 0)$ as the truncated version by only keeping the first $i$ coordinates. We also set $x_{\leq 0} = x$. Then we have the following properties of $F$.

**Lemma A.1.** *Let* $F(x) = \pi_{v^*} h_1(x) + \pi_{w^*} h_2(x)$ *for* $h_1, h_2$ *defined by* (14),(15). *Then we have the following:*

*(1).* $F(0) - \inf_x F(x) \leq \Delta_0 d$ *for some constant* $\Delta_0 > 0$.

*(2).* $\|\nabla h_i(x)\|_\infty \leq 23$ *and* $\|\nabla h_i(x)\| \leq 23\sqrt{d}, i = 1, 2$.

*(3).* $F(x)$ *is* $l_1$-*smooth for some constant* $l_1 > 0$.

*(4).* *If* $\text{prog}_1(x) < d$, $\|\nabla F(x)\| \geq 1$.

*(5).* $[\nabla h_i(x)]_{\leq \text{prog}_{\frac{1}{2}}(x)} = [\nabla h_i(x_{\leq \text{prog}_{\frac{1}{2}}(x)})]_{\leq \text{prog}_{\frac{1}{2}}(x)}, i = 1, 2$.

*(6).* *If* $\text{prog}_0(x)$ *is odd,* $\text{prog}_0(\nabla h_1(x)) \leq \text{prog}_{\frac{1}{2}}(x)$, $\text{prog}_0(\nabla h_2(x)) \leq \text{prog}_{\frac{1}{2}}(x) + 1$. *If* $\text{prog}_0(x)$ *is even,* $\text{prog}_0(\nabla h_1(x)) \leq \text{prog}_{\frac{1}{2}}(x) + 1$, $\text{prog}_0(\nabla h_2(x)) \leq \text{prog}_{\frac{1}{2}}(x)$.

*(7).* *If* $\text{prog}_{\frac{1}{2}}(x)$ *is odd,* $\nabla h_1(x) = \nabla h_1(x_{\leq \text{prog}_{\frac{1}{2}}(x)})$, $\nabla h_2(x) = \nabla h_2(x_{\leq 1 + \text{prog}_{\frac{1}{2}}(x)})$. *If* $\text{prog}_{\frac{1}{2}}(x)$ *is even,* $\nabla h_1(x) = \nabla h_1(x_{\leq 1 + \text{prog}_{\frac{1}{2}}(x)})$, $\nabla h_2(x) = \nabla h_2(x_{\leq \text{prog}_{\frac{1}{2}}(x)})$.

*Proof.* For Part (1), observing that $F(0) < 0$ and noting that $0 \leq \psi(u) \leq e, 0 \leq \phi(u) \leq \sqrt{2\pi e}$,

$$F(x) \geq -\psi(1)\phi([x]_1) - \sum_{i=2}^{d} \psi([x]_{i-1})\phi([x]_i) \geq -de\sqrt{2\pi e} = -d\Delta_0$$

with $\Delta_0 = e\sqrt{2\pi e}$, which completes its proof.

For Part (2), noting that $0 \leq \psi'(u) \leq \sqrt{54e^{-1}}$ and $0 \leq \phi'(u) \leq \sqrt{e}$, combining with the fact that for each $i = 1, 2$

$$\frac{\partial h_i}{\partial x_j}(x) \geq \psi(-[x]_{j-1})\phi'(-[x]_j) - \psi([x]_{j-1})\phi'([x]_j) - \psi'(-[x]_j)\phi(-[x]_{j+1}) - \psi'([x]_j)\phi([x]_{j+1})$$

yields

$$\left| \frac{\partial h_i}{\partial x_j}(x) \right| \leq e\sqrt{e} + \sqrt{54e^{-1}}\sqrt{2\pi e} \leq 23$$

implying $\|\nabla h_i(x)\| \leq 23$ and $\|\nabla h_i(x)\| \leq 23\sqrt{d}, \forall i = 1, 2$.

Parts (3) and (4) follow directly from (Carmon et al., 2020). Parts (5)-(7) follow from the observation that

$$\nabla h_1(x) = \nabla h_1([x]_1, \ldots, [x]_{2i+1}, 0, \ldots, 0), \text{ if } |x_{2j}| \leq \frac{1}{2}, \forall j \geq i+1$$

$$\nabla h_2(x) = \nabla h_2([x]_1, \ldots, [x]_{2i}, 0, \ldots, 0), \text{ if } |x_{2j-1}| \leq \frac{1}{2}, \forall j \geq i+1.$$

$\square$

## A.3. Lower bound for finite-state Markov chains

According to Part (6) of Lemma A.1, the above constructive $F$ and $f$ as (16) satisfy the conditions (12) and (13). Consider the following $F^*$

$$F^*(x) := \frac{L\lambda^2}{l_1} F\left(\frac{x}{\lambda}\right), \quad \text{where } \lambda = \frac{2l_1}{L}\epsilon. \tag{17}$$

Accordingly, we have

$$g^*(x;s) := \nabla f^*(x;s) = \frac{L\lambda}{l_1} \nabla f\left(\frac{x}{\lambda};s\right) = 2\epsilon \nabla f\left(\frac{x}{\lambda};s\right).$$

We note that

$$\nabla^2 F^*(x) = \frac{L}{l_1} \nabla^2 F\left(\frac{x}{\lambda}\right)$$

which implies that $F^*$ is $L$-smooth by Part (3) of Lemma A.1, and similarly we conclude $f^*(\cdot;s)$ is $L$-smooth for any $s \in \mathcal{S}$. Moreover, by Part (1) of Lemma A.1 we obtain that

$$F^*(0) - \inf_x F^*(x) = \frac{4l_1\epsilon^2}{L}(F(0) - \inf_x F(x)) \leq \frac{4l_1\Delta_0\epsilon^2}{L}d$$

and further we note for any $s \in \mathcal{S}$ and any $x$

$$\|\nabla f(x;s) - \nabla F(x)\|^2 \leq 2(\|h_1(x)\|^2 + \|\nabla h_2(x)\|^2) \leq 2 \cdot (23)^2 d$$

where we use Part (2) of Lemma A.1, implying that

$$\|g^*(x;s) - \nabla F^*(x)\|^2 \leq 8 \cdot 23^2 \epsilon^2 d.$$

Therefore by setting

$$d = \min\left\{\left\lfloor \frac{L\Delta}{4l_1\Delta_0\epsilon^2} \right\rfloor, \left\lfloor \frac{\sigma^2}{8 \cdot 23^2\epsilon^2} \right\rfloor\right\} \tag{18}$$

we guarantee that

$$F^*(0) - \inf_x F^*(x) \leq \Delta$$
$$\|\nabla F^*(x) - \nabla F^*(y)\| \leq L\|x - y\|, \ \forall x, y$$
$$\|\nabla f^*(x;s) - \nabla f^*(y;s)\| \leq L\|x - y\|, \ \forall x, y$$
$$\|g^*(x;s) - \nabla F^*(x)\|^2 \leq \sigma^2$$

which hence indicates that $F^* \in \mathcal{F}'(\Delta, L)$ and Assumption 4.1 is satisfied.

Further combining with (16) and Part (6) of Lemma A.1 yields that conditions (12) and (13) hold for $F^*$ and $f^*$, where we use that $\text{prog}_{\frac{1}{2}}(x) \leq \text{prog}_0(x), \forall x$. Therefore, for $B = 1$,

$$\max_{m \in M} \max_{l \leq t} \text{prog}_0(x_{l,m}^{\mathcal{A}[O_{F^*}]}) < d, \ \forall t \leq \frac{1}{2}\tau d \tag{19}$$

For any algorithm $\mathcal{A}$ with $B \geq 1$, we observe that by the construction of $F^*$ and $f^*$ and the Markov chain $P^*$, for any $t \geq 0$ and $m \in [M]$, there exists an algorithm $\tilde{\mathcal{A}}$ with $B = 1$ for which $\text{prog}_0(x_{t,m}^{\mathcal{A}[O_{F^*}]}) \leq \text{prog}_0(x_{t,m}^{\tilde{\mathcal{A}}[O_{F^*}]})$, since multiple samples do not contribute to additional progress of $x$ (This also proves the finite-state case of Lemma 5.3). Therefore, (19) also holds for $B \geq 1$, which implies that for any $m \in [M]$

$$\|\nabla F^*(x_{t,m}^{\mathcal{A}[O_{F^*}]})\| = \frac{L\lambda}{l_1}\left\|\nabla F\left(\frac{x}{\lambda}\right)\right\| \geq \frac{L\lambda}{l_1} \geq 2\epsilon, \ \forall t \leq \frac{1}{2}\tau d$$

where we use Part (4) of Lemma A.1 and note that $\text{prog}_1(x) \leq \text{prog}_0(x)$. Thus we conclude

$$N_{s,fi}^\epsilon(M, \Delta, L, \sigma^2, \tau) \geq \frac{1}{2}\tau d \geq \frac{\tau}{\epsilon^2}\min\left\{\left\lfloor \frac{L\Delta}{2l_1\Delta_0} \right\rfloor, \left\lfloor \frac{\sigma^2}{4 \cdot 23^2} \right\rfloor\right\}$$

completing the proof of Theorem 4.2.

### A.4. Lower bound for countable-state Markov chains

Note that Assumption 2.2 for the countable-state case is more general than Assumption 4.1 for the finite-case, which provides us with more flexibility on the construction of the "hard" example. In particular, we slightly modify the construction of the Markov chain $P^*$ in Section A.1 by further splitting states $v^*$ and $w^*$ into two substates, i.e., $v^* = \{v_1^*, v_2^*\}$ and $w^* = \{w_1^*, w_2^*\}$ such that conditioning on $v^*$ (or $w^*$) is visited, the probability that $v_1^*$ or $(w_1^*)$ is sampled is $q \in (0, 1)$. We denote the modified construction as $\tilde{P}^*$ Figure 2 visualizes our construction.

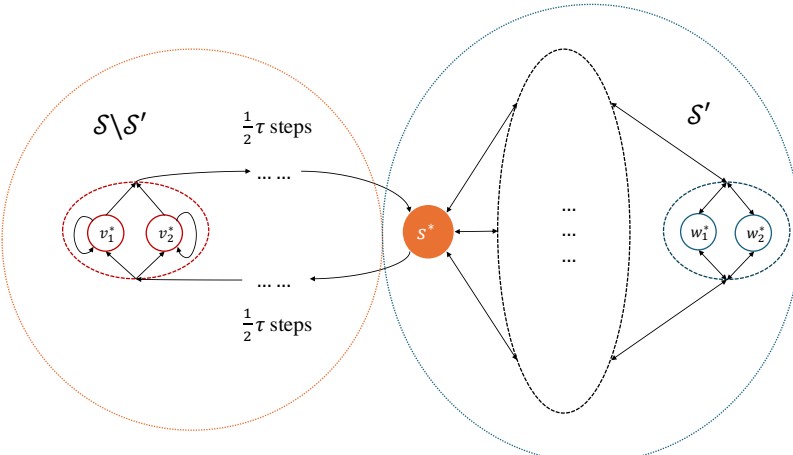

Figure 2. Construction of Markov chain $\tilde{P}^*$ with splitting states

We define $\pi_{v^*} = \pi_{v_1^*} + \pi_{v_2^*}$ and $\pi_{w^*} = \pi_{w_1^*} + \pi_{w_2^*}$ as the marginal probabilities of virtual states $v^*$ and $w^*$. Then, we construct our $F$ and $f$ for countable-states Markov chains as following: for each $i$-th coordinate of $f$

$$\text{if } s \in \{v_1^*, v_2^*\}, \; [g(x;s)]_i = [\nabla h_1(x)]_i \cdot \left(1 + \mathbb{1}\{i > \text{prog}_{\frac{1}{2}}(x)\}\left(\frac{\mathbb{1}_{s=v_1^*}}{q} - 1\right)\right),$$

$$\text{if } s \in \{w_1^*, w_2^*\}, \; [g(x;s)]_i = [\nabla h_2(x)]_i \cdot \left(1 + \mathbb{1}\{i > \text{prog}_{\frac{1}{2}}(x)\}\left(\frac{\mathbb{1}_{s=w_1^*}}{q} - 1\right)\right),$$

$$\text{otherwise, } f(x;s) = 0, \tag{20}$$

where $\mathbb{P}(s = v_1^* \mid s \in \{v_1^*, v_2^*\}) = \mathbb{P}(s = w_1^* \mid s \in \{w_1^*, w_2^*\}) = q \in (0, 1)$ and let $F(x) = \pi_{v^*} h_1(x) + \pi_{w^*} h_2(x)$. Then we have the following lemma.

**Lemma A.2.** *Considering stochastic gradient $g(x; s)$ constructed as* (20)*, the following statements hold:*

*(1). For $s \in \{v_1^*, v_2^*, w_1^*, w_2^*\}$, with probability at least $1 - q$, $\text{prog}_0(g(x;s)) \leq \text{prog}_{\frac{1}{2}}(x)$ and $g(x;s) = g(x_{\leq \text{prog}_{\frac{1}{2}}(x)}; s)$ for all $x$.*

*(2). For $s \notin \{v_1^*, v_2^*, w_1^*, w_2^*\}$, with probability 1, $\text{prog}_0(g(x;s_t)) \leq \text{prog}_{\frac{1}{2}}(x)$ and $g(x;s_t) = g(x_{\leq \text{prog}_{\frac{1}{2}}(x)}; s_t)$ for all $x$.*

*(3). For any $s$, with probability 1, $\text{prog}_0(g(x;s)) \leq 1 + \text{prog}_{\frac{1}{2}}(x)$ and $g(x;s) = g(x_{\leq 1 + \text{prog}_{\frac{1}{2}}(x)}; s)$ for all $x$.*

*(4). $\mathbb{E}_{s \sim \pi}[g(x;s)] = \nabla F(x)$.*

*Proof.* We firstly show Part (3). Note that by (20) and Part (7) of Lemma A.1, for any $x, s$, $[g(x;s_t)]_i = 0, \forall i > 1 + \text{prog}_{\frac{1}{2}}(x)$ in the sense that $[\nabla h_1(x)]_i = [\nabla h_2(x)]_i = 0, \forall i > 1 + \text{prog}_{\frac{1}{2}}(x)$, which implies $\text{prog}_0(g(x;s)) \leq 1 + \text{prog}_{\frac{1}{2}}(x)$. Moreover, by Part (7) of Lemma A.1, defining $x' := x_{\leq 1 + \text{prog}_{\frac{1}{2}}(x)}$ gives $\nabla h_1(x) = \nabla h_1(x')$ and $\nabla h_2(x) = \nabla h_2(x')$. Thus, we obtain $g(x;s) = g(x';s)$ for any $x, s$, implying Part (3).

For Part (1), we note that when $i \geq 1+\text{prog}_{\frac{1}{2}}(x)$ and $s \in \{v_2^*, w_2^*\}$, $g(x; s) = [\nabla h_j(x)]_{\leq \text{prog}_{\frac{1}{2}}(x)}$ for $j = 1, 2$, which implies $\text{prog}_0(g(x; s)) \leq \text{prog}_{\frac{1}{2}}(x), \forall s \in \{v_2^*, w_2^*\}$. Further, according to (5) of Lemma A.1, we have $g(x; s) = g(x_{\leq \text{prog}_{\frac{1}{2}}(x)}; s)$ for $s \in \{v_2^*, w_2^*\}$ and all $x$. Since $P(z = 0) = 1 - q$, hence Part (1) is proved.

Part (2) holds trivially in the sense that $g(x; s_t) = 0$ when $s \notin \{v_1^*, v_2^*, w_1^*, w_2^*\}$. Finally, Part (4) holds since $\mathbb{E}[\mathbb{1}_s/q \mid s \in \{v_1^*, v_2^*\}] = \mathbb{E}[\mathbb{1}_s/q \mid s \in \{w_1^*, w_2^*\}] = 1$. $\quad\square$

Also, we show in the following lemma that $g$ has bounded variance.

**Lemma A.3.** *For $F(x) = \pi_{v^*} h_1(x) + \pi_{w^*} h_2(x)$ and $g$ defined as (20), then for any Markov chain with stationary distribution $\Pi$, given any $x \in \mathbb{R}^d$, $t \geq 0$ and any initial distribution of the chain,*

$$\mathbb{E}\|g(x; s_t) - \nabla F(x)\|^2 \leq a_1 d + a_2 \frac{1-q}{q}$$

*for some constant $a_1, a_2 > 0$.*

*Proof.* By Part (4) of Lemma A.2, we know $\mathbb{E}_{s \sim \pi}[g(x; s)] = \nabla F(x)$.

Denote $i^* = 1 + \text{prog}_{\frac{1}{2}}(x)$. For any $s \in \{v_1^*, v_2^*, w_1^*, w_2^*\}$, we have

$$g(x; s) - \nabla F(x) = (0, \ldots, 0, [\nabla h_1(x)]_{i^*}(\mathbb{1}_{s=v_1^*}/q - 1), 0, \ldots, 0) + (1 - \tilde{\pi}_{v^*})\nabla h_1(x) - \tilde{\pi}_{w^*}\nabla h_2(x), \text{ if } s \in \{v_1^*, v_2^*\}$$
$$g(x; s) - \nabla F(x) = (0, \ldots, 0, [\nabla h_2(x)]_{i^*}(\mathbb{1}_{s=w_1^*}/q - 1), 0, \ldots, 0) + (1 - \tilde{\pi}_{w^*})\nabla h_2(x) - \tilde{\pi}_{v^*}\nabla h_1(x), \text{ if } s \in \{w_1^*, w_2^*\}.$$

When $i^* - 1$ is odd, from Part (6) of Lemma A.1 we know that $[\nabla h_1(x)]_{i^*} = 0$. Therefore,

$$\|g(x; s) - \nabla F(x)\|^2 \leq 2\|\nabla h_1(x)\|^2 + 2\|\nabla h_2(x)\|^2 \leq 4 \cdot 23^2 d, \quad s \in \{v_1^*, v_2^*\}$$

$$\|g(x; s) - \nabla F(x)\|^2 \leq 3|[\nabla h_2(x)]_{i^*}|^2(\mathbb{1}_{s=w_1^*}/q - 1)^2 + 3\|\nabla h_1(x)\|^2 + 3\|\nabla h_2(x)\|^2$$
$$\leq 3 \cdot 23^2(\mathbb{1}_{s=w_1^*}/q - 1)^2 + 6 \cdot 23^2 d, \quad s \in \{w_1^*, w_2^*\}$$

and

$$\|g(x; s_t) - \nabla F(x)\|^2 = \|\nabla F(x)\|^2 \leq 4 \cdot 23^2 d, \quad \text{when } s \notin \{v_1^*, v_2^*, w_1^*, w_2^*\}$$

where we use (2) of Lemma A.1. Combining the above three inequalities, it yields that when $i^* - 1$ is odd, for any Markov chain, any $x$, $t \geq 0$ and any initial distribution of the chain,

$$\mathbb{E}\|g(x; s_t) - \nabla F(x)\|^2 \leq a_1 d + a_2 \frac{1-q}{q}$$

where $a_1 = 6 \cdot 23^2$, $a_2 = 3 \cdot 23^2$ and we use that $\mathbb{E}[(\mathbb{1}_s/q - 1)^2 \mid s \in \{w_1^*, w_2^*\}] = (1 - q)/q$. The case when $i^* - 1$ is even can be derived similarly. $\quad\square$

Then, we are ready to show Lemma 5.2. Again we first focus on the case of $B = 1$ and then generalize it to $B \geq 1$.

*Proof of Lemma 5.2.* We construct $F^*$ the same as (17) with $\lambda = 2l_1\epsilon/L$ and

$$g^*(x; s) = \frac{L\lambda}{l_1} g\left(\frac{x}{\lambda}, s\right)$$

with $g(x; s)$ defined as (20). Then by Lemma A.3, we have

$$\mathbb{E}\|g^*(x; s_t) - \nabla F^*(x)\|^2 \leq 4a_1 d\epsilon^2 + \frac{4a_2(1-q)}{q}\epsilon^2, \quad \forall t \geq 0.$$

Then, define

$$B_t := \mathbb{1}\left\{\exists x : \text{prog}_0(g^*(x; s_t)) = 1 + \text{prog}_{\frac{1}{2}}(x)\right\}.$$

Note that under the construction of the Markov chain $\tilde{P}^*$ and $F^*$ and $g^*$, for any zero-respecting algorithm $A$

$$B_{t+k} = 0, \ \forall k = 1, \ldots, \frac{1}{2}\tau, \ \text{conditioning on } B_t = 1.$$

That is to say within every $\frac{1}{2}\tau$ iterations $B_t$ can be 1 at most once. And Part (1) of Lemma A.2 indicates that the probability of $B_t$ being 1 is no greater than $q$. Let $k(t) := \max_{m\in[M]} \max_{l\le t} \text{prog}_0(x_{l,m}^{\mathcal{A}[O]_{F^*}})$. Then, the above implies that

$$k(t) \le \sum_{l\le t} B_l$$

which then gives

$$\mathbb{P}(k(t) \ge d) \le \mathbb{P}\left(\sum_{l\le t} B_l \ge d\right)$$

$$= \mathbb{P}\left(\exp\left(\sum_{l\le t} B_l\right) \ge e^d\right)$$

$$\le e^{-d}\mathbb{E}[e^{\sum_{l\le t} B_l}]$$

$$\le e^{-d}\mathbb{E}[e^{\sum_{i=1}^{\lceil 2t/\tau\rceil} z_i}]$$

$$= e^{-d}(1 - q + eq)^{\lceil 2t/\tau\rceil}$$

$$\le e^{\lceil 4t/\tau\rceil q - d}$$

where $z_i$s are i.i.d. Bernoulli random variable with succeeding probability at most $q$ and in the fourth inequality we use the fact that the number of $B_l$ that can possibly be 1 is at most $\lceil 2t/\tau\rceil$ and they are independent if not forcing to be zero. Therefore, we conclude that for any $\delta \in (0,1)$ and $q \in (0,1)$ with probability at least $1 - \delta$,

$$k(t) < d, \ \forall t \le \frac{\tau(d - \log(1/\delta))}{4q}$$

which completes the proof of Lemma 5.2. Similarly we can use the same technique in the last section to show the countable-state case of Lemma 5.3.

Now to show Theorem 3.1, setting

$$d = \min\left\{\left\lfloor\frac{L\Delta}{4l_1\Delta_0\epsilon^2}\right\rfloor, \left\lfloor\frac{\sigma^2}{8a_1\epsilon^2}\right\rfloor\right\} \tag{21}$$

and

$$\frac{1}{q} = 1 + \frac{\sigma^2}{8a_2\epsilon^2} \tag{22}$$

yields that $F^* \in \mathcal{F}(\Delta, L)$ and Assumption 2.2 is satisfied. By Part (4) of Lemma A.1 and Lemma 5.2, choosing $\delta = 1/2$ renders that for any $m \in [M]$ with probability at least $1/2$,

$$\|\nabla F^*(x_{t,m}^{\mathcal{A}[O_{F^*}]})\| \ge 2\epsilon, \ \forall t \le \frac{\tau(d-1)}{4q}$$

which implies that

$$\mathbb{E}\|\nabla F^*(x_{t,m}^{\mathcal{A}[O_{F^*}]})\| \ge \epsilon, \ \forall t \le \frac{\tau(d-1)}{4q}.$$

Therefore, we conclude that

$$N_s^\epsilon(M, \Delta, L, \sigma^2, \tau) \ge \frac{\tau(d-1)}{4q} \succeq \frac{\tau\sigma^2}{\epsilon^2} + \frac{\tau\sigma^2}{\epsilon^2}\min\{c_1\sigma^2, c_2 L\Delta\}$$

by the selections of $d, q$ as (21),(22) for some constants $c_1, c_2 > 0$.

# B. Convergence Analysis of MaC-SAGE

In the following, we denote $\pi_i$ the stationary probability according to state $i$.

**Definition B.1.** Define $t_{mix}(\epsilon) := \inf\{l \geq 1 \mid d_{TV}(\mu P^l, \Pi) \leq \epsilon\}$. Then $\tau_{mix} = t_{mix}(1/4)$ is the mixing time of the chain.

**Lemma B.2.** *We have the following claims:*

- $d_{TV}(\mu P^{t+1}, \Pi) \leq d_{TV}(\mu P^t, \Pi)$.

- *For $k \geq 2$, $t_{mix}(2^{-k}) \leq (k-1)\tau_{mix}$.*

- *Moreover,*

$$\sum_{k=0}^{T} d_{TV}(\mu P^k, \Pi) \leq c_0 \tau_{mix}, \quad \forall T \geq 0$$

*for some $c_0 > 0$.*

*Proof.* The first two claims are directly from (Levin & Peres, 2017).

To see the third claim, we note that

$$\sum_{k=0}^{T} d_{TV}(\mu P^k, \Pi) \leq \sum_{k=0}^{\infty} d_{TV}(\mu P^k, \Pi)$$

$$\leq \sum_{l=0}^{\tau_{mix}} d_{TV}(\mu P^l, \Pi) + \sum_{k=0}^{\infty} \sum_{l=t_{mix}(2^{-k})+1}^{t_{mix}(2^{-(k+1)})} d_{TV}(\mu P^l, \pi)$$

$$\leq d_{TV}(\mu, \Pi)\tau_{mix} + \sum_{k=2}^{\infty} (t_{mix}(2^{-(k+1)}) - t_{mix}(2^{-k}))2^{-k}$$

$$\leq d_{TV}(\mu, \Pi)\tau_{mix} + \sum_{k=2}^{\infty} k2^{-k}\tau_{mix}$$

$$\leq d_{TV}(\mu, \Pi)\tau_{mix} + 2\tau_{mix}$$

which completes the proof with $c_0 = d_{TV}(\mu, \pi) + 2$. □

**Theorem B.3.** *Let $\mathcal{S}$ be the state space of the Markov chain with $|\mathcal{S}| = N$. Consider any real-valued mapping $v : \mathcal{S} \to \mathbb{R}^{Nd}$ and $\mu$ is any initial distribution. Without particularly claiming, $v(\cdot)$ is a row vector. Define $V = [v(1)^T, \ldots, v(N)^T]^T \in \mathbb{R}^{N \times Nd}$. Then*

$$\mathbb{E}_\mu \left( \frac{1}{T} \sum_{t=0}^{T-1} v(s_t) - \Pi^T V \right) = \frac{1}{T} \sum_{t=0}^{T-1} \mu^T(P^t - \mathbf{1}\Pi^T)V$$

$$T\mathbb{E}_\Pi \left\| \frac{1}{T} \sum_{t=0}^{T-1} v(s_t) - \Pi^T V \right\|^2 \leq \max_i \|v(i)\|_\infty^2 \|I - \mathbf{1}\Pi^T\|_\infty + 2c_0 \max_i \|v(i)\|_\infty^2 \tau_{mix}$$

$$T\mathbb{E}_\mu \left\| \frac{1}{T} \sum_{t=0}^{T-1} v(s_t) - \Pi^T V \right\|^2 \leq T\mathbb{E}_\Pi \left\| \frac{1}{T} \sum_{t=0}^{T-1} v(s_t) - \Pi^T V \right\|^2 + 3c_0 \max_i \|g(i)\|_\infty^2 \tau_{mix}$$

*where $\mathbb{E}_\mu(\cdot)$ means the initial state $s_0$ follows $\mu$; $g(i) = v(i) - \Pi^T V$.*

*Proof.* We firstly show the first equality. Note that

$$\mathbb{E}_\mu \left( \frac{1}{T} \sum_{t=0}^{T-1} v(s_t) - \Pi^T f \right) = \frac{1}{T} \sum_{k=0}^{T-1} (\mu^T P^k V - \Pi^T V)$$

$$= \frac{1}{T} \sum_{k=0}^{T-1} \mu^T (P^k - \mathbf{1}\Pi^T) V$$

where we observe that $\mu^T \mathbf{1} = 1$.

Then we turn to show the second inequality. By definition, we have

$$T\mathbb{E}_\Pi \left\| \frac{1}{T} \sum_{t=0}^{T-1} v(s_t) - \Pi^T V \right\|^2 = \mathbb{E}_\Pi \| v(s_0) - \Pi^T V \|^2$$

$$+ \frac{2}{T} \sum_{k=1}^{T-1} (T-k) \mathbb{E}_\Pi [(v(s_0) - \Pi^T V)(v(s_k) - \Pi^T V)^T]. \quad (23)$$

Moreover,

$$\mathbb{E}_\Pi [(v(s_0) - \pi^T V)(v(s_k) - \pi^T V)^T] = \sum_i \pi_i v(i) \sum_j [P^k]_{i,j} v(j)^T - \sum_{i,j} \pi_i \pi_j v(i) v(j)^T$$

$$= \sum_i \pi_i v(i) \sum_j [P^k - \mathbf{1}\Pi^T]_{i,j} v(j)^T$$

$$\leq \sum_i \pi_i \| v(i) \|_\infty \sum_j |[P^k - \mathbf{1}\Pi^T]_{i,j}| \| v(j) \|_\infty$$

$$\leq \max_i \| v(i) \|_\infty^2 \| P^k - \mathbf{1}\Pi^T \|_\infty$$

Applying Lemma B.2 to (23) yields

$$T\mathbb{E}_\Pi \left( \frac{1}{T} \sum_{t=0}^{T-1} v(s_t) - \Pi^T V \right)^2 \leq \max_i \| v(i) \|_\infty^2 \| I - \mathbf{1}\Pi^T \|_\infty + 2c_0 \max_i \| v(i) \|_\infty^2 \tau_{mix}$$

which completes the proof of the second inequality.

To obtain the third inequality, defining $g(i) = v(i) - \Pi^T V$ we aim to bound

$$T \left| \mathbb{E}_\mu \left\| \frac{1}{T} \sum_{k=0}^{T-1} g(s_k) \right\|^2 - \mathbb{E}_\Pi \left\| \frac{1}{T} \sum_{k=0}^{T-1} g(s_k) \right\|^2 \right|$$

$$\leq \left| \frac{1}{T} \sum_{k=0}^{T-1} \mathbb{E}_\mu \| g(s_k) \|^2 - \mathbb{E}_\Pi \| g(s_k) \|^2 \right| + \frac{2}{T} \sum_{k=0}^{T-1} \sum_{l=k+1}^{T-1} \left| \mathbb{E}_\mu (g(s_k) g(s_l)^T) - \mathbb{E}_\pi (g(s_k) g(s_l)^T) \right|.$$

Note

$$\left| \mathbb{E}_\mu (g(s_k) g(s_l)^T) - \mathbb{E}_\pi (g(s_k) g(s_l)^T) \right| = \left| \sum_{i,j} \mu_i g(j) ((P^k)_{i,j} - \pi_j) \sum_r ((P^{l-k})_{j,r} - \pi_r) g(r)^T \right|$$

$$= \left| \sum_{i,j} \mu_i g(j) ((P^k - \mathbf{1}\Pi^T))_{i,j} \sum_r ((P^{l-k} - \mathbf{1}\Pi^T))_{j,r} g(r)^T \right|$$

$$\leq \max_i \| g(i) \|_\infty^2 \| P^l - \mathbf{1}\Pi^T \|_\infty.$$

Thus,

$$T \left| \mathbb{E}_\mu \left\| \frac{1}{T} \sum_{k=0}^{T-1} g(s_k) \right\|^2 - \mathbb{E}_\Pi \left\| \frac{1}{T} \sum_{k=0}^{T-1} g(s_k) \right\|^2 \right|$$

$$\leq \frac{1}{T} \sum_{k=0}^{T-1} \| \mu^T (P^k - \mathbf{1}\Pi^T) \|_\infty \max_i \| g(i) \|_\infty^2 + \frac{2}{T} c_0 \max_i \| g(i) \|_\infty^2 \sum_{k=0}^{T-1} \tau_{mix}$$

$$\leq \frac{1}{T} \sum_{k=0}^{T-1} \| \mu^T (P^k - \mathbf{1}\Pi^T) \|_\infty \max_i \| g(i) \|_\infty^2 + 2 c_0 \max_i \| g(i) \|_\infty^2 \tau_{mix}$$

$$\leq c_0 \max_i \| g(i) \|_\infty^2 \tau_{mix} (2 + T^{-1}).$$

Combining all the above completes the proof. $\qquad \square$

**Corollary B.4.**

$$\mathbb{E} \left\| \sum_{i=1}^n (y_t^i - \pi_i)(\nabla f_i(x_t) - \nabla F(x_t)) \right\|^2 = \mathcal{O} \left( \frac{\sigma^2 \tau_{mix}}{t} \right)$$

$$\mathbb{E} \left\| \sum_{i=1}^n (y_t^i - \pi_i) \right\|^2 = \mathcal{O} \left( \frac{\sigma^2 \tau_{mix}}{t} \right)$$

*Proof.* For any $t$, setting $v(i) = [0, \ldots, \nabla f_i(x_t)^T - \nabla F(x_t)^T, \ldots, 0]$ and noting $\| v(i) \|_\infty^2 \leq \| \nabla f_i(x_t) - \nabla F(x_t) \|^2 \leq \sigma^2$ completes the first result. Similarly setting $v(i) = [0, \ldots, 0, 1, 0, \ldots, 0]$ completes the second. $\qquad \square$

**Theorem B.5.** *Considering MaC-SAGE (Algorithm 1) and supposing all conditions in Theorem 4.4 hold, then*

$$\mathbb{E} \left[ \min_{t<T} \| \nabla F(x_t) \|^2 \right] = \mathcal{O} \left( \frac{\tau L \Delta}{T} \right) + \tilde{\mathcal{O}} \left( \frac{\sigma^2 \max\{\tau, \tau_{mix}\}}{T} \right).$$

*Proof.* Using the smoothness of $F$,

$$F(x_{t+1}) - F(x_t) \leq \langle \nabla F(x_t), x_{t+1} - x_t \rangle + \frac{L}{2} \| x_{t+1} - x_t \|^2$$

$$= -\gamma_t \langle \nabla F(x_t), G_t \rangle + \frac{L \gamma_t^2}{2} \| G_t \|^2$$

$$= \frac{\gamma_t}{2} \| \nabla F(x_t) - G_t \|^2 - \frac{\gamma_t}{2} \| \nabla F(x_t) \|^2 - \frac{\gamma_t}{2} (1 - L\gamma_t) \| G_t \|^2$$

$$\leq \frac{\gamma_t}{2} \| \nabla F(x_t) - G_t \|^2 - \frac{\gamma_t}{2} \| \nabla F(x_t) \|^2 - \frac{\gamma_t}{4} \| G_t \|^2$$

for $\gamma_t \leq 1/(2L)$.

Note that by $a_i(t) = \sup\{l \geq 1 \mid l \leq t, s_l = i\}$

$$\| \nabla F(x_t) - G_t \|^2 \leq 2 \left\| \sum_{i=1}^n (y_t^i - \pi_i) \nabla f_i(x_t) \right\|^2 + 2 \left\| \sum_{i=1}^n y_t^i (\nabla f_i(x_{a_i(t)}) - \nabla f_i(x_t)) \right\|^2$$

$$= 2 \left\| \sum_{i=1}^n (y_t^i - \pi_i)(\nabla f_i(x_t) - \nabla F(x_t)) \right\|^2 + 2 \left\| \sum_{i=1}^n y_t^i (\nabla f_i(x_{a_i(t)}) - \nabla f_i(x_t)) \right\|^2$$

$$\leq 2 \left\| \sum_{i=1}^n (y_t^i - \pi_i)(\nabla f_i(x_t) - \nabla F(x_t)) \right\|^2 + 2 \sum_{i=1}^n y_t^i \| \nabla f_i(x_{a_i(t)}) - \nabla f_i(x_t) \|^2$$

$$\leq 2 \left\| \sum_{i=1}^n (y_t^i - \pi_i)(\nabla f_i(x_t) - \nabla F(x_t)) \right\|^2 + 2 \sum_{i=1}^n y_t^i L^2 \| x_{a_i(t)} - x_t \|^2$$

where the third inequality follows the convexity of $l_2$-norm by $\sum_{i=1}^{n} y_t^i = 1$ and $y_t^i \geq 0$ by definition; the last one follows $L$-smoothness of each $f_i$. Further we have

$$\|x_t - x_{a_i(t)}\|^2 \leq \sum_{l=a_i(t)}^{t-1} (t - a_i(t))\gamma_l^2 \|G_l\|^2.$$

Denoting $e_t := \left\|\sum_{i=1}^{n} (y_t^i - \pi_i)(\nabla f_i(x_t) - \nabla F(x_t))\right\|^2$, we obtain

$$\gamma_t \|\nabla F(x_t) - G_t\|^2 \leq 2\gamma_t e_t + 2L^2 \gamma_t \sum_{i=1}^{n} \sum_{l=a_i(t)}^{t-1} y_t^i(t - a_i(t))\gamma_l^2 \|G_l\|^2$$

$$\leq 2\gamma_t e_t + L \sum_{i=1}^{n} y_t^i \sum_{l=a_i(t)}^{t-1} \gamma_l^2 \|G_l\|^2$$

when $\gamma_t \leq \frac{1}{2L \max_i \{t - a_i(t)\}}$. Further note that

$$\sum_{t=0}^{T-1} \sum_{l=a_i(t)}^{t-1} \gamma_l^2 \|G_l\|^2 = \sum_{t=0}^{T-1} \gamma_t^2 \|G_t\|^2 (b_i(t) - t - 1)$$

where $b_i(t) = \inf\{l \mid l > t, \ s_l = i\}$. Thus, by letting $\Delta_T = F(x_0) - F(x_T)$ we have

$$\sum_{t=0}^{T-1} \frac{\gamma_t}{2} \|\nabla F(x_t)\| \leq \Delta + \sum_{t=0}^{T-1} \gamma_t e_t - \sum_{t=0}^{T-1} \frac{\gamma_t}{4} \|G_t\|^2 + \frac{L}{2} \sum_{i=1}^{n} \sum_{t=0}^{T-1} y_t^i \gamma_t^2 \|G_t\|^2 (b_i(t) - t - 1).$$

Moreover, since

$$\mathbb{E}\left[\sum_{i=1}^{n} \sum_{t=0}^{T-1} y_t^i \gamma_t^2 \|G_t\|^2 (b_i(t) - t - 1) \mid \mathcal{F}_t\right] = \sum_{i=1}^{n} \sum_{t=0}^{T-1} y_t^i \gamma_t^2 \|G_t\|^2 \mathbb{E}[b_i(t) - t - 1 \mid \mathcal{F}_t]$$

$$\leq \sum_{i=1}^{n} \sum_{t=0}^{T-1} y_t^i \gamma_t^2 \|G_t\|^2 \tau_{hit}$$

$$= \tau_{hit} \sum_{t=0}^{T-1} \gamma_t^2 \|G_t\|^2$$

$$\leq \sum_{t=0}^{T-1} \frac{\gamma_t}{4L} \|G_t\|^2$$

for $\gamma_t = \frac{1}{4L(\max_i\{t - a_i(t)\} + \tau_{hit})} \leq \frac{1}{4L\tau_{hit}}$, it yields

$$\mathbb{E}\left[\min_{t<T} \|\nabla F(x_t)\|^2\right] \leq \mathbb{E}\left[\frac{2\Delta_T}{\sum_{t=0}^{T-1} \gamma_t}\right] + \mathbb{E}\left[\frac{2\sum_{t=0}^{T-1} \gamma_t e_t}{\sum_{t=0}^{T-1} \gamma_t}\right].$$

Noting that according to Lemma A.5 of (Even, 2023)

$$\frac{T}{\sum_{t=0}^{T-1} \gamma_t} \leq \frac{1}{T} \sum_{t=0}^{T-1} \gamma_t^{-1} \leq 8L \log(n)\tau_{hit}$$

and by Corollary B.4

$$\mathbb{E}\left[\sum_{t=0}^{T-1} \gamma_t e_t\right] \leq \frac{1}{4L\tau_{hit}} \mathbb{E}\left[\sum_{t=0}^{T-1} e_t\right] = \mathcal{O}\left(\frac{\sigma^2 \tau_{mix} \log T}{L\tau_{hit}}\right).$$

Finally combining with the fact $\tau_{hit} \leq \tau$ we obtain

$$\mathbb{E}\left[\min_{t<T} \|\nabla F(x_t)\|^2\right] = \mathcal{O}\left(\frac{\tau L\Delta}{T}\right) + \tilde{\mathcal{O}}\left(\frac{\sigma^2 \max\{\tau, \tau_{mix}\}}{T}\right)$$

$\square$

