# OpenReview forum: "Improved Lower Bounds for First-order Stochastic Non-convex Optimization under Markov Sampling"
_ICML.cc/2025/Conference — ICML 2025 poster_

### Official Review · Reviewer_FyNE · 2025-03-13

**Overall Recommendation:** 4

**Summary:**

This paper studies non-convex stochastic optimization when the data is generated from a Markov chain. This is unlike most papers on the topic where one usually assumes that the noise process affecting the gradients is an i.i.d. process. The goal of this paper is to establish information-theoretic lower bounds on the number of samples needed to achieve an $\epsilon$-accurate stationary point (in expectation). For both countable and finite-state Markov chains, the paper provides tighter lower bounds than those existing. For the latter case, an SVRG-like algorithm is proposed that additionally maintains estimates of the entries of the stationary distribution of the underlying Markov chain. It is shown that this algorithm achieves the $O(1/\epsilon^2)$ rate for this setting.

**Claims And Evidence:**

The claims made in the paper are all rigorously supported by detailed proofs.

**Essential References Not Discussed:**

Relevant references are all well cited and discussed.

**Experimental Designs Or Analyses:**

There are no experiments in this paper. This is by no means a limitation since the focus of the paper is to establish fundamental lower bounds.

**Methods And Evaluation Criteria:**

This is a theoretical paper, and there is not much to evaluate here.

**Other Comments Or Suggestions:**

In the finite-state setting, I have a comment regarding the MaC-SAGE algorithm. The algorithm needs to keep track of the number of occurrences of each state. I was wondering if this can be avoided using the following idea.

Suppose the mixing time of the underlying Markov chain is $\tau$. Then, the subsampled sequence $s_0, s_{\tau}, s_{2 \tau}, \ldots, ..$ is ``almost" i.i.d. with high probability. Now suppose one uses exactly the same algorithm as one would in the i.i.d. case, but updates it once in every $\tau$ time-steps. In this way, one effectively runs the algorithm on i.i.d. data. So one should expect pretty much the same guarantees as in the i.i.d. case, inflated by a delay of $\tau$, since one now effectively uses $T/\tau$ samples, where $T$ is the total number of samples. Isn't this going to recover the same guarantees as MaC-SAGE, since it appears from Theorem 4.4. that the final error-bound is the i.i.d. bound scaled by $\tau$?

More generally, unless I am mistaken, given data from an ergodic Markov chain, one could run any optimization algorithm (with no modification) on a subsampled data set (where the subsampling gap is informed by the mixing time of the Markov chain) and achieve the same guarantees as in the i.i.d. case, with $T$ replaced by the number of effective samples $T/\tau$.

Some discussion on this matter would be very helpful. Essentially, I want to understand whether there is any need to develop new optimization algorithms for the Markov setting, or would subsampling suffice.

**Other Strengths And Weaknesses:**

Strengths
---------------
- Time-correlated Markovian data shows up in a variety of stochastic approximation problems. Compared to their i.i.d. counterparts, much less is understood for such settings. In this regard, I find the contribution of the paper significant in improving the understanding of what is fundamentally achievable in the non-convex setting.

- While the MaC-SAGE algorithm is essentially very similar to variance-reduced algorithms like SAG, SAGA, and SVRG, I still find it very useful to know that such an algorithm is minimax optimal in the finite-state Markov setting.

- I think the ideas used for proving upper and lower bounds in this paper can find much broader applicability beyond just non-convex optimization.

There are no particular weaknesses that I can find.

**Questions For Authors:**

I have some clarifying questions. I am happy to raise my score once they have been addressed.

- Q1) The assumption of bounded variance in Assumption 2.2. seems a bit weird to me. In particular, at any given time $t$, the distribution of the state $s_t$ has not yet converged to the stationary distribution. Thus, $\nabla F(x)$ is not the expected value of $g(x; s_t)$. Isn't this correct? If yes,  $\mathbb{E} \Vert g(x; s_t) - \nabla F(x) \Vert^2_2$ is not really the variance of $g(x; s_t)$ (unlike the i.i.d. case). What does it mean to assume a uniform bound then on this expectation? Wouldn't this bound depend on $x$, more generally, in the Markov case? Perhaps the authors can use the TD learning example presented earlier to validate this assumption.

- Q2) How is the step-size sequence chosen to arrive at the result in Theorem 4.4.? This does not seem to be specified in the statement of Theorem 4.4., nor in the description of MaC-SAGE. Does it require knowledge of the mixing time $\tau$? If yes, it brings me back to my earlier comment of simply sub-sampling the data based on the knowledge of $\tau$.

- Q3) The lower bounds pertain to the number of samples needed to ensure that the *expected* value of the gradient is below a specified tolerance. Can similar lower bounds be derived if one seeks guarantees with high-probability? To be more precise, given a failure probability $\delta$, suppose we wish to determine the minimum number of samples $N(\delta, \epsilon)$ needed to ensure that $\Vert \nabla F(x_t) \Vert $ is below $\epsilon$. Is it possible to characterize $N(\delta, \epsilon)$ using the techniques developed in this paper?

**Relation To Broader Scientific Literature:**

The main contribution of the paper refines the existing literature on Markovian stochastic optimization by deriving tighter lower bounds than those available previously. The authors do a good job of positioning their contributions in this context.

**Theoretical Claims:**

I skimmed through the main proof ideas and they appear correct to me.

---

> ### Author Rebuttal · Authors · 2025-03-31
>
> **Comments on subsampling**: We really appreciate the reviewer for initiating such an interesting discussion. In the following we briefly present our understanding and hopefully would provide some insights to the comments. We think the reviewer’s intuition is correct and is aligned with ours. However, we would like to emphasize that simply using a subsampled sequence might not be enough to achieve the same rate as MaC-SAGE. This is because even under the i.i.d. case, vanilla SGD suffers from a slower rate. In other words, to get a faster rate as MaC-SAGE, variance-reduced techniques are needed. For the finite-state case, the main difference between the Markov sampling and uniform sampling (i.e., finite-sum for the i.i.d. case considered in literature) lies in that the stationary distribution $\Pi$ is unknown and non-uniform, which makes its estimation necessary before applying existing algorithms designed for the i.i.d. case. Once it is guaranteed that the estimate of the stationary distribution converges no slower than the algorithm’s rate in the i.i.d. case, one can directly apply algorithms for the i.i.d. case (e.g. SAG, SVRG) to the Markovian case with the final rate scaled by $\tau$. That is also the idea behind MaC-SAGE, where we maintain $y_t$ as an estimate of $\Pi$. Also please refer to lines 300-310 on the right column for discussions. Finally, it is worth noting that the analysis for the Markovian case is non-trivial because of the correlation among time-dependent samples, although algorithmically the rate seems similar (up to $\tau$) to the i.i.d. case.
>
> **Q1**: We thank the reviewer for bringing this to our attention. Actually, we are able to weaken such a strong assumption by now taking the expectation according to the stationary distribution $\Pi$, i.e., $\mathbb{E}_{s \sim \Pi}\Vert g(x;s) - \nabla F(x) \Vert \le \sigma^2$, which is then similar to the bounded variance in i.i.d. setting. We note that considering this modified assumption does not affect our lower bound results (i.e., Theorem 3.1 still holds), as the proof of Lemma A.3 holds for any distribution of the chain.
>
> **Q2**: The stepsize $\gamma_t$ is chosen as line 1026, which only needs the knowledge of hitting time. We will add its expression to the main text in our updated version.
>
> **Q3**: We think our proof techniques could be used to obtain high-probability results. Specifically, in Appendix A.3, eq. (19) holds almost surely, which hence indicates line 652 holds almost surely. Therefore, Theorem 4.2 is valid with probability one. For Theorem 3.1, similarly, we have shown Lemma 5.2, which is a high-probability result (see line 799 for details). In the proof, we set $\delta = 0.5$ to get the final expectation bound (see lines 810-814). Thus, if we leave $\delta$ as a parameter, a high-probability version can be characterized.

---

> > ### Comment · Reviewer_FyNE · 2025-04-03
> >
> > Thank you for addressing my comments.
> >
> > I am satisfied with the rebuttal and increase my score to '4'.

---

### Official Review · Reviewer_dSkt · 2025-03-13

**Overall Recommendation:** 3

**Summary:**

This paper studies the sample complexity of stochastic optimization for smooth, non-convex functions when the noise variables form a Markov chain instead of being i.i.d. The authors obtain a lower bound of $\Omega{\tau\epsilon^{-4}}$ for stationary Markov processes with a countable state space, where $\tau$ is the hitting time of the Markov chain, and a lower bound of  $\Omega{\tau\epsilon^{-2}}$ for finite-state Markov chains, where $\tau$ is the hitting time of the Markov chain. A new algorithm, MaC-SAGE, which nearly matches the lower bound in case of finite-state Markov chains. The work provides theoretical complexity bounds and an efficient algorithm for optimization with Markovian sampling.

**Claims And Evidence:**

All statements and theorems given by the authors in the main text are proved, the proofs are given in the appendix.

**Essential References Not Discussed:**

No essential references are missed.

**Experimental Designs Or Analyses:**

Not applicable.

**Methods And Evaluation Criteria:**

The analysis of sample complexity in first-order stochastic optimization is well-established. The authors provide an algorithm-independent lower bound on sample complexity for non-convex functions with markovian noise and, in the case of a finite-state Markov chain, propose an algorithm that nearly matches this bound.

**Other Comments Or Suggestions:**

The paper contains a few typos:

1. Line 595-596: should be $\|h_i(x)\|_{\infty}$ in the first norm.

2. In Section 2.3 it is written that $g(x;s) := \nabla f(x,s)$ and then the authors use one or the other notation in the text, e.g., in Assumption 4.1 or in formula (20) in the supplementary material. Different notations for the same object make it difficult to understand the text.

3. In Lemma A.2, somewhere there is an index $t$ in $s$ and somewhere there is not;

4. In Assumption 4.1 it is written that $\|g(x,s_t) - \nabla F(x)\| \leq \sigma^2$. Is it an expectation missing or we rely on an almost sure bound here?

**Other Strengths And Weaknesses:**

The provides an improvement in lower bounds for non-convex optimization under Markovian sampling. Its originality lies in tightening existing bounds and proposing the MaC-SAGE algorithm, which nearly matches the new lower bound for finite-state space Markov chain. However, the paper considers only discrete Markov noise, while in [1] an upper bound is given for non-convex functions and an arbitrary ergodic Markov chain admitting a stationary distribution. Also in [2] the uniformly ergodic Markov noise is considered and the lower bound is given for strongly convex functions. At the same time, is not clear to me, where exactly the current construction relies on the fact that we are working with a finite state space in the proofs of Section 4.2. In particular, factors relating $\tau_{mix}$ and $\tau_{hit}$ might depend on the size of the space $|S|$.

[1] Arjevani, Yossi, et al. "Lower bounds for non-convex stochastic optimization." Mathematical Programming 199.1 (2023): 165-214.

[2] Beznosikov, Aleksandr, et al. "First order methods with markovian noise: from acceleration to variational inequalities." Advances in Neural Information Processing Systems 36 (2023): 44820-44835.

**Questions For Authors:**

Where exactly the current construction relies on the fact that we are working with a finite state space in the proofs of Theorem 4.2? If there are some hidden constants depending on the cardinality of the space where the Markov chain runs, I suggest to trace them explicitly and provide them in the statement.

**Relation To Broader Scientific Literature:**

This paper builds on existing research in stochastic optimization by strengthening lower bounds for nonconvex optimization under Markov sampling, improving on the work of [2] by establishing a tighter $\Omega(\tau \varepsilon^{-4})$ bound that matches the upper bound obtained in [1]. This extends previous results for independent noise [3] to a more complex Markovian setting where dependencies between samples affect the convergence rate. In addition, the authors refine the lower bound for a finite Markov chain $\Omega(\tau \varepsilon^{-2})$ and propose a new MaC-SAGE algorithm that tags the obtained lower bound.

[1] Dorfman, Ron, and Kfir Yehuda Levy. "Adapting to mixing time in stochastic optimization with markovian data." International Conference on Machine Learning. PMLR, 2022.

[2] Even, Mathieu. "Stochastic gradient descent under Markovian sampling schemes." International Conference on Machine Learning. PMLR, 2023.

[3] Arjevani, Yossi, et al. "Lower bounds for non-convex stochastic optimization." Mathematical Programming 199.1 (2023): 165-214.

**Theoretical Claims:**

1. In the proof of Theorem B.3., the transition from
$$
\sum_{k=1}^{T-1}\|P-e\Pi^T\|_{\infty} \leq c_0 \tau_{mix}
$$
is not explained in the second and third claims (see line 912). To me it should be an additional factor $|S|$ coming from
$$
\sum_{k=1}^{T-1} \|P^{k}- \1 \Pi^T\|_{\infty},
$$
since the supremum might be attained on different rows of $P^{k}$ for different indices $k$. If this is not the case, please provide nore detailed derivation. Otherwise I do not see how this fact directly follows Lemma B.2.

2. In Corollary B.4. the second inequality has a dependence on $\sigma$, but $\||v_i\|_{\infty} = 1$ in this case.

---

> ### Author Rebuttal · Authors · 2025-03-31
>
> ## Response to "Theoretical Claims"
>
> 1. The proof of Theorem B.3: We think the proof of Theorem B.3 is correct and we will add details in the updated version. We detailedly explain how line 912 is derived as follows:
>
> First, we derive line 907 from line 906:
> Denoting $v_{max} := \max_i \Vert v(i) \Vert_{\infty}$, we have
> $$
> \begin{aligned}
> \sum_i \pi_i \Vert v(i) \Vert_{\infty} \sum_j \vert P^k - \mathbf{1}\Pi^T\vert_{i,j}  \Vert v(j) \Vert_{\infty}
> &\le \sum_i \pi_i \Vert v(i) \Vert_{\infty} \sum_j \vert P^k - \mathbf{1}\Pi^T\vert_{i,j}  \max_{j} \Vert v(j) \Vert_{\infty} \\
> &\le \sum_i \pi_i \max_i \Vert v(i) \Vert_{\infty} \sum_j \vert P^k - \mathbf{1}\Pi^T\vert_{i,j}  \max_j \Vert v(j) \Vert_{\infty} \\
> &= v_{\max}^2 \sum_i \pi_i \sum_j \vert P^k - \mathbf{1}\Pi^T\vert_{i,j} .
> \end{aligned}
> $$
> where due to display issue, we use $\vert A \vert_{i,j}$ to denote $|a_{ij}|$.
>
>
> Since by definition of the matrix infinity norm (i.e., for matrix $A$, $\Vert A \Vert_{\infty} := \max_i \sum_j |a_{ij}|$), for any $i$, $\sum_j  | P - \mathbf{1}\Pi^T \mid_{i,j}  \le \Vert P - 1\Pi^T \Vert_{\infty}$, then we conclude line 907 by further noting $v_{max}^2 = \max_i \Vert v(i) \Vert_{\infty}^2$ and $\sum_i \pi_i = 1$.
>
> Then, note that for finite-state case, since Lemma B.2 holds for any initial distribution $\mu$, setting $\mu = 1_{s_0 = i}$, meaning the chain starts from state $i$ with probability one, it yields that
> $$
> 	\sum_{k=0}^T \max_i d_{TV}(P^k(\cdot \mid s_0 = i), \Pi) \le c_0 \tau_{mix}.
> $$
> Moreover, by definition of total variation distance, i.e., $d_{TV}(p,q) := (1/2) \sum_{z}|p(z) - q(z)|$, we conclude $\Vert P^k - \mathbf{1}\Pi^T \Vert_{\infty} = 2 \max_{i} d_{TV}(P^k(\cdot \mid s_0=i), \Pi)$. Combining all these facts gives the second term of line 912.
>
> To see the first term of line 912, which is the bound for the first term in (23), we simply set $k=0$ for line 900, which hence is bounded by $\max_i \Vert v(i) \Vert_{\infty}^2 \Vert I - \mathbf{1}\Pi^T \Vert_{\infty}$.
>
> 2. Typo in Corollary B.4: We apologize for that typo. The second bound in Corollary B.4 should not depend on $\sigma$ and we will fix it in the updated version.
>
>
> ##  Response to "Other Strengths And Weaknesses"
>
> **About the cardinality of finite state space on the results**: We note that the hitting time in Theorem 4.2 is usually a function of the cardinality of the state space in practice. That is to say the number of states affects the sample complexity by means of the hitting time captured in our theorem. How the cardinality relates to hitting time varies case by case.
>
> ## Response to "Other Comments Or Suggestions"
>
> 1. Typo on line 595: We apologize for the typo. We will fix it in the updated version.
>
> 2. Notation of $g$: In the paper, we consistently define $g(x;s) = \nabla f(x;s)$, i.e., $g$ is the first-order stochastic gradient sampling from the Markov chain. In the updated version, we will clarify such notations for easy understanding.
>
> 3. Notation in Lemma A.2: We will drop all subscripts of $t$ in the updated version.
>
> 4. About Assumption 4.1: There is no expectation on the norm, saying the bound is assumed to hold almost surely. That also distinguishes the construction for the finite-state case from the countable-state case. Please refer to lines 282-292 for detailed discussion.
>
> ## Response to "Questions for Authors"
>
> We note that in our construction, the cardinality of the finite Markov chain is $\Omega(\tau)$ due to the fact that there are at least $\tau$ states between $v^*$ and $s^*$ (see Figure 1). Since the hitting time of the chain is usually a function of the cardinality, our bound hence implicitly depends on the state space cardinality.

---

### Official Review · Reviewer_VvVo · 2025-03-14

**Overall Recommendation:** 4

**Summary:**

The paper proves a lower complexity bound of $ O(\tau_{mix} \varepsilon^{-4}) $ for smooth, non-convex stochastic optimization under Markovian noise with countable states. For finite state space, a lower complexity bound  $ O( \varepsilon^{-2}) $ is also given, and a proposed method to match the lower bound to logarithmic factors is proposed.

**Claims And Evidence:**

The analysis well supports the claims. The lower bound improves from the previous result of Even 2023 under the same setup. The lower bound is also tight, in the sense of matching the upper bound in prior works.

**Essential References Not Discussed:**

Related works are relatively complete.

**Experimental Designs Or Analyses:**

Not applicable.

**Methods And Evaluation Criteria:**

No numerical studies are presented.

**Other Comments Or Suggestions:**

See above.

**Other Strengths And Weaknesses:**

Overall, I think the paper makes good contributions to the broad stochastic optimization field, where many ML methods operate under Markov chains. The results are presented clearly, and the proofs are intuitive to understand.
The consideration of finite state space is a good addition to the results, and the proposed algorithm, although similar to the classical variance reduced method for non-convex problems, yields strong convergence rate in the markovian noise setting.

However, the writing might not be friendly enough. For example, I am a little confused about the definition of zero-respecting algorithms in section 2.3. It seems this algorithm class is general, but what are the exceptions?

Additionally, this work focus on smooth functions, I wonder what's the results with more generic assumption, i.e., bounded gradients?

**Questions For Authors:**

See above.

**Relation To Broader Scientific Literature:**

The works add an important piece for stochastic optimization under the non-convex setting.

**Theoretical Claims:**

The proof is not checked carefully. The proof seems to follow similar works in optimization and lower complexity bound analysis.

---

> ### Author Rebuttal · Authors · 2025-03-31
>
> We sincerely thank the reviewer for positive and encouraging comments on our paper. In our updated version, we will promote the writing to make it clearer and easier to follow for the readers.
>
> **For zero-respecting algorithms we consider in the paper**, we note that zero-respecting algorithms require the initial point $x_0$ to be zero, i.e., $x_0 = 0$ due to (5). That is to say random initialization is not allowed for zero-respecting algorithms. However, it is worth noting that such limitation on the initialization is also seen in lower bound analysis of literature [Even’23, Arjevani et al’23, Beznosikov et al’24, Duchi et al’12] and it does not affect the convergence result of the algorithm. We are willing to generalize our results to randomized algorithms in the future.
>
> **About extension to bounded gradient assumption**: We appreciate the reviewer’s suggestion. Extension to bounded gradient case is interesting and we hope to address it in the future. Also we would like to note that smoothness is a common assumption for convergence analysis in optimization literature [Ghadimi and Lan’13, Even’23, Roy et al’22, Dorfman and Levy’22]. Therefore, our results are applicable for optimization problems under standard assumptions.

---

### Official Review · Reviewer_eLLJ · 2025-03-14

**Overall Recommendation:** 4

**Summary:**

This paper studies the lower bound of sample complexity of general first-order algorithms for stochastic non-convex optimization problems under Markov sampling. They first show that for samples drawn from a stationary Markov chain with countable state space, the sample complexity is at least $\Omega(\epsilon^{-4})$. Moreover, for finite-state Markov chains, they show a $\Omega(\epsilon^{-2})$ lower bound of the sample complexity and propose a new algorithm that is proved to (nearly) match the lower bound.

**Claims And Evidence:**

The claims are supported by clear and convincing evidence. The lower bounds are derived via carefully constructed functions and Markov chains, while the algorithm’s analysis rigorously addresses Markovian sampling.

**Essential References Not Discussed:**

None.

**Experimental Designs Or Analyses:**

There is no experiment.

**Methods And Evaluation Criteria:**

The construction of the hard instance is standard in the literature. The design of the near-optimal algorithm seems new and interesting to me.

**Other Comments Or Suggestions:**

1. Page 3, Left, Equation (2): It is better to explicitly give the definition of $g(\cdot)$
2. Page 3, Left, line 151: Is it $\bar{s}_{t}$?
3. Page 3, Right, line 163: It is better to give the dimension of $x_{t, i}$ and $x_t$
4. Page 4, Left, line 185: What is support?
5. Page 4, Left, line 205: Is $x_{t+1/2}$ used in the iteration?
6. Page 4, Definition 2: It seems to me $\tau_w$ should be defined with $\inf$
7. Page 5, Equation (7): Should it be $|S_T(\mathcal{A})|$?
8. Page 5: Line 239: Can you explain more on When $N^{\epsilon}_{s}(M, \Delta, L, \sigma^2, \tau)$ is lower bounded by $N$,

i.e., $N_{s}(M, \Delta, L, \sigma^2, \tau) \geq N_T$

9. Why the sample complexity of Algorithm 1 is $O(\tau \epsilon^{-2})$?
10. The introduction of the algorithm class can have more explanation.

**Other Strengths And Weaknesses:**

None.

**Questions For Authors:**

1. In terms of the sample complexity gap, what is the core difference in analysis between the infinite-state Markov chain and finite-state Markov chain?
2. Why consider different oracles in the infinite-state Markov chain and finite-state Markov chain?
3. What is the key difference when constructing the hard instance under this Markov sample scheme?
4. How does the number of states affect the performance of MaC-SAGE？

**Relation To Broader Scientific Literature:**

As stated in the paper, the previous lower bound under this setting is $\mathcal{\Omega}(\tau \epsilon^{-1})$. This paper improves it to  $\mathcal{\Omega}(\tau \epsilon^{-4})$. I think it is a huge improvement and helps us understand the boundary of the first-order algorithms.

**Theoretical Claims:**

I did not check the proof line by line, but the technique is standard.

---

> ### Author Rebuttal · Authors · 2025-03-31
>
> 1.  By using $:=$, we actually define $g(\theta; s, s’) := (\phi(s)^T \theta - r(s, s’) - \gamma \phi(s’)^T \theta)\phi(s)$.
>
> 2. We will fix it in the updated version.
>
> 3. We thank the reviewer for the suggestion. By our notation, we mean $x_{t,i}$ being the $i$-th point at $t$-th iteration, whose dimension is $d$. We denote $x_t$ just for a collection of all updated points $x_{t,1},\dots, x_{t,M}$. And when $M=1$, $x_t = x_{t,1}$. The reason we define such multiple points of query is to include a broader class of algorithms, e.g., momentum-based algorithms. We present the example of Randomized ExtraGradient in line 204, where $M=2$, meaning two points are maintained to update at every iteration. In the updated version, we will clarify further clarify it.
>
> 4. In our setting, the support of a vector means the collection of all non-zero coordinates, i.e., $support(x)= \\{ i : x[i] \ne 0, \forall 1 \le i \le d \\}$.
>
> 5. We apologize for the typo. $x_{t+1/2}$ should be used in the update of $x_{t+1}$ in line 206. We will fix it in the updated version.
>
> 6. We apologize for the typo of missing $\inf$. We will fix it in the updated version.
>
> 7. We will fix it in the updated version.
>
> 8. By the definition of $N_s^{\epsilon}$, to ensure $N_s^{\epsilon} \ge N_T$ with $N_T$ being some constant, we only need to guarantee 1) the existence of an oracle (due to $\sup$ over $O_s$); 2) the existence of a function (due to $\sup$ over $\mathcal{F}$); 3) for any algorithm $\mathcal{A}$ (due to $\inf$ over $\mathbf{A}_{zr}$), 4) the smallest number of samples used by $\mathcal{A}$ causes its output leading to the expectation of the gradient norm firstly lies below the level of $\epsilon$.
>
> 9. Denote $\tilde{x}_T$ as the point which realizes the minimization, i.e., $\tilde{x}_T \in arg \min E\Vert \nabla F(x_t) \Vert^2$. Then, in order to force $\mathbb{E}\Vert \tilde{x}_T \Vert \le \epsilon$, it suffices to force $\mathbb{E}\Vert \tilde{x}_T \Vert^2 \le \epsilon^2$. Setting the bound on the right hand side of Theorem 4.4 gives that $T = \tilde{O}(\tau \epsilon^{-2})$. Moreover, since only one sample is drawn at each iteration, it concludes the sample complexity of Algorithm 1 is $\tilde{O}(\tau \epsilon^{-2})$.
>
> 10. We appreciate the reviewer’s suggestion. We will provide more detailed explanations of the algorithm class in the updated version. Roughly speaking, the algorithm class we consider in the paper captures almost all popular first-order methods in literature. At every iteration, the algorithm is allowed to take all histories of previous samples and $ x_0,\dots, x_t$ to generate $x_{t+1}$ following (5) which is satisfied by almost all first-order methods in literature. Note that (5) generalizes [Even’23] where $x_{t+1}$ is linearly spanned by previous points and sampled gradients.
>
> **Q1**. The core difference lies in how the stochastic gradient $g$ is constructed in the two settings. Assumption 2.2 for countable-state chains is weaker than Assumption 4.1 for the finite case (see lines 282–292). To address this, we design a countable-state Markov chain (Figure 2) by splitting $v^*, w^*$ into substates $v_1^*, v_2^*, w_1^*, w_2^*$ and define $g$ via (20). Unlike the finite case (Figure 1), where the transition from the parent of $v^*$ to $v^*$ happens with probability one, in the countable case, $v_1^*, v_2^*$ share the same parent, and the transition splits probabilistically: to $v_1^*$ with probability $q$, and to $v_2^*$ with $1-q$. Equivalently, the chain transitions to $v^*$, then flips a coin to select $v_1^*$ or $v_2^*$. This added randomness, combined with (20), makes the event “$prog_0(x)$ increases by one” succeed with probability $q$ (as shown in Lemma 5.2). As a result, more samples are needed—compared to the finite case—to ensure $prog_0(x) = d$ and hence $\Vert \nabla F(x) \Vert$ is sufficiently small. See lines 333–359 for details.
>
> **Q2**. The main reason of considering different oracles is to bridge the gap between our lower bounds and upper bounds for different settings. Essentially we are searching for tight bounds under different settings. While we could get $\epsilon^{-4}$ for the finite case under a more general Assumption 2.2, we aim to prove that an improved bound $\epsilon^{-2}$ is tight under a more specific/easier setting for the finite state space case. This is both motivated by practical consideration and also the existing upper bounds. Then we propose a new algorithm such that the sample complexity can be improved to $\epsilon^{-2}$ under such a stricter but practical Markov-chain class further with the matching lower bound.
>
> **Q3**. Please refer to Q1.
>
> **Q4**. We note the hitting time is a function of the number of states which is different for different Markov chains. Therefore, the convergence rate of MaC-SAGE is affected case by case.

---

### Official Review · Reviewer_UhUm · 2025-03-18

**Overall Recommendation:** 2

**Summary:**

This paper presents sample complexity lower bounds for stochastic gradient descent under a Markovian sampling assumption. In particular, there are two theorems in the paper showing lower bounds $\Omega(\epsilon^{-4})$ for Markov chains with countably infinite state space and $\Omega(\epsilon^{-2})$ for finite state space chains. The results also depend on a constat $\tau$, which is an upper bound on the Markov chain's hitting time.

In addition, the paper presents a new algorithm called MaC-SAGE for finite state space chains whose sample complexity upper bound matches the lower bound in the paper.

**Claims And Evidence:**

See Theoritical Claims section.

**Essential References Not Discussed:**

The authors are encouraged to compare their results (on the upper bound) with the results in the following papers on stochastic optimization with dependent data:

[1] William Powell, Hanbaek Lyu, "Stochastic optimization with arbitrary recurrent data sampling", ICML 2024

[2] Ahmet Alacaoglu and Hanbaek Lyu, "Convergence of First-Order Methods for Constrained Nonconvex Optimization with Dependent Data", ICML 2023

Especially, Thm. 3.8 in [1] seems to establish the matching upper bound $O(\tau/T)$, where in fact \tau is replaced by the random target time, a quantity that is generally smaller than the hitting time. Please also see my question on the upper bound in the theory section.

**Experimental Designs Or Analyses:**

N/A

**Methods And Evaluation Criteria:**

N/A

**Other Comments Or Suggestions:**

I am reserving my recommendation due to the (1) question/gap in the proof of lower bound and (2) novelty of the proposed algorithm and upper bound. I am willing to revisit my score if my concerns are successfully addressed.

**Other Strengths And Weaknesses:**

**Strengths:**

1. The paper presents two new lower-bounds for SGD with Markovian sampling which were not available in the literature. For the countable state space case, the dependence on $\epsilon$ matches that of i.i.d. sampling. The dependence of the lower bound on $\tau$ demonstrates the additional difficulty of the problem for poorly behaved Markov chains. The bounds for non-convex problems are much tighter than previous results.

2. The version of SAG in Algorithm 1 has sample complexity matching the lower-bound for finite state chains. It also works by maintaining an approximation of the stationary measure which is interesting and new to my knowledge.

**Weaknesses:**

1. More detail in some of the proofs would also be helpful. The construction of the Markov chains for the lower bounds seems a somewhat vague. Especially the for the countably infinite case on page 13, a more explicit definition of the transition kernel would be helpful. Defining the transition to states $v_1^*$ and $w_1^*$ conditional on being in $\{v_1^*, v_2^*\}$ and $\{w_1^*, w_2^*\}$ as is done below (20) is a bit confusing to me.

2. I don't understand how the Markovian dependence of the data sampling is handled in the proofs. For instance, I am unsure how we reach the conclusion (19) on page 12 in the proof of Theorem 4.2. It i also not clear to me how the construction on page 10 guarantees the hitting time is bounded above by $\tau$. See also the comments in the Theoretical claims section.

**Questions For Authors:**

1. There are two separate lower bounds for countably infinite and finite state space chains. However, the proofs dont seem to rely on the cardinality of the state space. It seems like both constructions can be done with a finite state space with size depending on $\tau$. The key driver of the difference in the two theorems seems to be the strengthening of the bounded variance assumption. Why are the theorems separated based on infinite vs. finite state space?

2. Also, for a countably infinite state space the assumption of bounded hitting time seems very strong. I think the stationary measure would have to be supported on a finite subset of $S$ for this to hold, which effectively reduces to the finite state space case. Can another assumption be used for countably infinite state space?

3. The paper mentions a number of times that the Markov chains considered are assumed to be stationary. Why is this necessary?

**Relation To Broader Scientific Literature:**

The paper's main contributions are lower bounds for SGD with Markovian sampling and non-convex objectives. The prior known lowerbound of [Even, 2023] was of the order $\Omega(\epsilon^{-1})$. This paper tightens this to $\Omega(\epsilon^{-4})$ for Markov chains on countably infinite state space and $\Omega(\epsilon^{-2})$ for finite state space. This is on par with similar results for i.i.d. sampling (e.g. [Arjevani et al. 2023]).

**Theoretical Claims:**

I have read through the proofs of Theorems 3.1, 4.1, and 4.4. I have some concerns on the correctness of their argument for the main results.

1. I think the convergence rate in Theorem 4.4 may depend on the mixing time as well. The last line of the convergence proof of MaC-SAGE on  page 19 asserts that $\tau_{\text{mix}} \preceq \tau_{\text{hit}}$ citing [Levin & Peres, 2017]. My understanding is that its is not true in general that the mixing time can be bounded by the hitting time. For such a bound to hold, Theorem 10.22 in [Levin & Peres, 2017] requires reversibility of the Markov chain and some holding probability ($P(x, x) \geq 1/2$) to ensure aperiodicity.

2. The proof of Theorem 3.1 (lines 780-793), the authors seem to argue that the sum $\sum_{l\le t} B_{l}$ of the indicators $B_{l}$ that progress was made at time $l$ can be written as the sum of i.i.d. Bernoully random variable, only based on the fact that the there is at most one 1 between a time interval of length $\tau/2$. I don't think this is true since $B_{l}$'s are defined according to the trajectory of a Markov chain. This part of the analysis must be fixed for me to recommend acceptance, since otherwise the validity of their key result, Thm. 3.1, the improved lower bound for Makovian first-order methods, is not justified.

3. Also, I have a question on their new algorithm MaC-SAGE and the upper bound $O(\tau/T)$ on the best-case expected gradient norm squared. The algorihtm seems to be a version of SAG (if not the same) in the Markovian setting, and [Even '23] already establishes a matching upper bound. Also the additional reference [Powell and Lyu '24] establishes a similar upper bound with random target time in place of the hitting time for regularized MISO ran on general recurrent data samples. This work also handles constrained nonconvex optimization. Compared to these existing results, I do not see a compelling reason to introduce a SAG-like algorithm if the only purpose is to obtain upper bound $O(\tau/T)$. What additional advantage does the proposed algorithm and the accompanying result provide? Does the proposed algorithm show competitive performance against these algorithms?

---

> ### Author Rebuttal · Authors · 2025-03-31
>
> **TC1**: After double-checking, we modified the theorem. Now the rate scales with $\max ( \tau_{mix},\tau_{hit})$, but note MaC-SAGE remains optimal (up to constants).
>
> **TC2**: We clarify that we are **not** claiming $B_l$s are i.i.d. Bernoulli r.v., but we claim $z_i$s (see its definition in the following) are i.i.d. Bernoulli r.v. We explain it as follows: First, we construct a Markov chain in which there are two states $v^*, w^*$ such that $\tau/2$ steps must be taken to commute each other. Then, we further introduce randomness in states $v^*, w^*$ to make sure that in the ideal case, when $\tau/2$ steps are taken, the event “$prog_0(x)$ increases by one” happens with probability at most $q$. To do this, we split $v^*$ (and $w^*$) into two substates $v_1^*, v_2^*$ (and $w_1^*, w_2^*$) and construct the gradient by (20). By Lemma A.2, every $\Omega(\tau)$ steps there is at most one $B_l$ being one, with its succeeding probability (if it could) at most $q$. To see line 790, in the ideal case, at least $\tau / 2$ steps are needed to increase $prog_0(x)$ by one and this becomes possibly true with probability at most $q$. That is to say, when $v^*$ or $w^*$ is visited, a coin is flipped where the flip’s result determines whether or not the event “$prog_0(x)$ increases by one” succeeds. And also the results of coin flips are independent across every time when $v^*$ or $w^*$ is visited. Therefore, $z_i$s record the results of coin flips, which are i.i.d.
>
> **TC3**: To highlight novelty: [Even’23] assumes uniform stationary distribution $\pi$ ($\pi_i = 1/n$), while [Powell & Lyu’24] require prior knowledge of $\pi$. In contrast, MaC-SAGE requires no information about $\pi$ and allows it to be non-uniform. We design $y_t$ to estimate $\pi$ and address this challenge.
>
> **W1**: We explain our construction of Markov chains by referring to Figures 1,2. Actually, transition probabilities are not critical for proving lower bounds; only the chain's structure matters.
> In the countable-state case (Figure 2), for the $S\setminus S’$ part (orange circle), starting from $v^*$ (union of $v_1^*, v_2^*$, red dashed circle), the chain can move only in one direction until hitting $s^*$; similarly for $s^*$ to $v^*$. The number of states along each path between $v^*$ and $s^*$ is $\tau/2$. The structure of the subchain in $S’$ (blue circle) is flexible—for example, it may be a complete graph. This construction ensures commuting between $v^*$ and $w^*$ requires at least $\tau/2$ steps, while maintaining ergodicity of the chain.
>
> The substates $v_1^*, v_2^*$ lie within $v^*$, and similarly for $w^*$. Figure 2 shows: 1) exactly one common parent state of $v_1^*, v_2^*$; 2) conditioned on the parent, transition to $v_1^*$ happens with probability $q$, and to $v_2^*$ with $1 - q$. It is equivalent that, upon reaching $v^*$, a coin is flipped to determine $v_1^*$ (w.p. $q$) or $v_2^*$ (w.p. $1-q$). The conditional probability below (20) defines this flip.
>
> **W2**: The constructed Markov chain and $f$ in (16) force any algorithm to iterate at least $\Omega(\tau)$ steps to make an increase in $prog_0(x)$. Then by Lemma 5.1, $\Omega(\tau d)$ samples are needed to make $\nabla F$ small (see lines 360-380). To see boundedness of hitting time, we note by our construction, as long as the hitting time of the subchain supported on $S’$ (blue circle) is $O(\tau)$, due to hitting time of the subchain in orange circle is $O(\tau)$, the hitting time of the whole chain is $O(\tau)$.
>
> **Q1**: We note our lower bounds rely on hitting time, which is a function of cardinality of (finite) state space. Due to space limit, please refer to responses to Q1&2 of Reviewer eLLJ about differences and reasons of separating Theorems 3.1&4.2.
>
> **Q2**: We acknowledge the limitation in finite hitting time assumption. We explain the idea how to construct a chain with bounded mixing time and hope to show it in future. First we design a cyclic-like subchain in orange circle with the number of states between $v^*$ and $s^*$ being $O(\sqrt{\tau})$. Second, we consider any subchain in blue circle whose mixing time is $O(\tau)$. If we restrict on the orange or blue subchain, we have the mixing times of them are $O(\tau)$. Then the mixing time of the composed chain is upper bounded by the larger mixing time of the orange or blue component. This is shown by [Madras&Randall’02] for reversible chains, while we conjecture it is also true for non-reversible chains.
>
> [1].Madras, Neal, and Dana Randall. Markov chain decomposition for convergence rate analysis. Annals of Applied Probability,2002.
>
> **Q3**: We acknowledge our proof techniques only suit stationary chains. But we note that many applications (e.g. TD learning, RL) belong to stationary case. And we highlight it is the first time improved lower bounds are provided for Markov sampling, with a new algorithm showing optimality of our bounds. Extending to non-stationary case is interesting and we hope to address it in future.

---

### Decision · Program_Chairs · 2025-05-01

**Decision:**

Accept (poster)

**Comment:**

The paper studies the sample complexity of stochastic optimization for smooth, non-convex functions when the noise variables form a Markov chain instead of being i.i.d.

In general, most of the reviewers support the paper. However, the are some concerns that the reviewer pointed out during the discussion:

1) If the lower bound (Thm. 4.2) still depends on the hitting time but the upper bound (Thm. 4.4) also depends on potentially much larger mixing time (e.g., for periodic chains), then clearly these two bounds do not match. Also, since Thm. 4.4 depends also on the mixing time but the upper bound on [Powell & Lyu’24] does not, so the author's claim about optimality up to a constant factor does not seem correct without further modification of the setting.

2) The reviewer was not convinced if this is a significant enough of a motivation to develop a new Markov chain optimization algorithm when there are well-established works out there. From the author's response, it seems that they wanted to additionally address the challenge of possibly not knowing the stationary distribution of the Markov chain. But does it fit the overall theme of the paper to proposing a lower bound for Markovian optimization algorithms and then providing a matching upper bound?

In many practical instance of MCMC, one has the known target distribution $\pi$ which is hard to sample from in i.i.d. manner (e.g., uniform distribution over a complicated set), one designs an MCMC that correctly converge to the target distribution so that one can at least asymptotically approximately sample from $\pi$. In such situation, there is no reason to additionally estimate the stationary distribution.

In cases when one has an MCMC sampling algorithm with unknown stationary distribution, and suppose for some reason we believe it has some meaningful stationary distribution with respect to which we want to solve some expected loss minimization. In such situations, the author's motivation of additionally adding estimation step may make sense. However, the authors uses basic approximation of the stationary distribution using the empirical distribution by recursively updating proportion of times that the chain spends at each state. If the Markov chain is ergodic, this empirical estimate converges as fast as the Markov chain mixes and the estimation bias will be exponentially small. So the novelty of the proposed MaC-SAGE algorithm seems incremental over the existing results.

The authors are expected to discuss all these points in the final version of the paper.